

# Plant controls on post-fire nitrogen availability in a pine savanna

Cari D. Ficken and Justin P. Wright

Department of Biology, Duke University, Durham, NC, 27708, USA

*Correspondence to*: Cari D. Ficken (cari.ficken@duke.edu)

Key words: nitrogen, mineralization, nitrification, fire, longleaf pine, ammonium, nitrate, plant resource uptake, $\delta^{15}N$

**Abstract.** Many ecosystems experience drastic changes to soil nutrient availability associated with fire, but the magnitude and duration of these changes are highly variable among vegetation and fire types. In pyrogenic pine savannas across the south eastern United States, pulses of soil inorganic nitrogen (N)

occur in tandem with ecosystem-scale nutrient losses from prescribed burns. Despite the importance of this management tool for restoring and maintaining fire-dependent plant communities, the contributions of different mechanisms underlying fire-associated changes to soil N availability remain unclear. Pulses of N availability following fire have been hypothesized to occur through (1) changes to microbial cycling rates and (2) direct ash deposition; we further hypothesize that (3) changes to plant sink strength

may contribute to ephemeral increases in soil N availability. Here, we document fire-associated changes to N availability across the growing season in a longleaf pine savanna in North Carolina. To differentiate between possible mechanisms driving soil N pulses, we measured net microbial cycling rates and changes to soil $\delta^{15}N$ before and after a burn. We found no evidence for changes in microbial activity, and limited evidence that ash deposition could account for the increase in ammonium

availability to more than 5-25 times background levels. We conclude that a temporary dampening of vegetation demand for N following fire may contribute to the observed increases in inorganic N availability.



## 1 Introduction

Temporal heterogeneity in resource supply is ubiquitous across ecosystems (Schimel and Bennett, 2004;James and Richards, 2006;Archer et al., 2014) and such resource pulses can be important if they contribute disproportionately to the overall resource budget of an ecosystem (McClain et al., 2003).

Because they vary in magnitude and frequency, nutrient pulses across ecosystems differ in their potential to influence community and ecosystem dynamics. Despite compelling modelling-based evidence suggesting that nutrient pulses can influence ecological dynamics including species richness (Tilman and Pacala, 1993), physiological nutrient uptake constraints (Bonachela et al., 2011), and stoichiometric coupling (Appling and Heffernan, 2014), it can be difficult to predict when and where

temporal nutrient heterogeneity will occur. This uncertainty makes it difficult to assess the conditions under which temporal heterogeneity in nutrient supply might influence community- or ecosystem-level functioning.

Nutrient dynamics in pyrogenic systems may be especially variable in time because fire is a major disturbance that influences nitrogen (N) availability (Wan et al., 2001). There is a consensus that, across

ecosystems, pulses of soil N, in particular ammonium ($NH_4^+$), occur in response to fire (Huber et al., 2013;Wan et al., 2001). However the duration and magnitude of these pulses vary strongly by fuel composition, and consequently among forest and fire types (Wan et al., 2001). In northern conifer forests, stand-replacing fires can result in increased soil $NH_4^+$ concentrations detectable more than one year following the burn (Smithwick et al., 2005;Turner et al., 2007). In contrast, in pine forests of the

south eastern US (e.g. longleaf pine savannas), where prescribed fires only consume the understory vegetation, $NH_4^+$ concentrations following fires are more variable; some studies have documented no change in soil N pools following fire (Christensen, 1977;Richter et al., 1982), while others have documented immediate increases (2 days) that quickly dissipate (Lavoie et al., 1992). This suggests a need for localized studies with ecosystem-specific temporal data resolution to evaluate the mechanism

behind changes in soil N availability following fire.

Pine savannas in the southeastern US are often managed with prescribed fires in the absence of recurring natural fires to maintain habitat for endangered species (Sorrie et al., 2006) and nutrient losses or pool redistributions from these fires can be substantial (Boring et al., 2004;Wilson et al.,



2002;Wilson et al., 1999). In addition to large quantities of carbon (C) released through fuel consumption (Boring et al., 2004), prescribed fires can release up to 50% of the phosphorus (P) and up to 75% of the N that was stored in the understory biomass (Carter and Foster, 2004;Wan et al., 2001). These nutrients can be lost through volatization, or redistributed as ash in low intensity fires. Despite

the ecosystem-level nutrient losses associated with fires, short-term pulses of increased N availability in the soil are also observed following prescribed forest fires across vegetation types (Certini, 2005;Schafer and Mack, 2010;Smithwick et al., 2005;Wan et al., 2001) including in longleaf pine (*Pinus palustris*) savannas (Boring et al., 2004). In longleaf pine savannas, because these nutrient pulses occur during a period of rapid post-fire plant regrowth, they may influence successional patterns

(Shenoy et al., 2013), plant diversity, and ecosystem productivity.

The mechanisms driving these ephemeral increases in N availability following fire remain poorly resolved, and so it remains difficult to predict how a specific fire will influence local N availability and turnover. Fire can decrease N availability if N is volatized and lost from the system in high intensity fires (Lavoie et al., 2010;Certini, 2005). On the other hand, fire can increase N availability if it spurs

microbial turnover of organic matter (Wilson et al., 2002;Certini, 2005), returns nutrient-rich ash to the system (Boring et al., 2004), or decreases the vegetation demand for N. Short term increases in soil N availability may not translate to longer-term ecosystem retention if N is lost through leaching or as gaseous products during turnover.

In addition to difficulties associated with assessing the relative importance of each mechanism

influencing post-fire N availability, logistical challenges remain to accurately measure N availability. First, changes to soil N availability are likely to occur rapidly following fire. Since microbial turnover occurs on a span of hours to days, and plants in fire-adapted systems begin re-sprouting within days to weeks, changes to N availability in pyrogenic systems are also likely to be ephemeral. Previous studies of post-fire N dynamics in longleaf pine savannas have relied on monthly or less-frequent soil samples

(Wilson et al., 2002;Lavoie et al., 2010), but this sampling resolution may be too coarse if changes in N dynamics occur rapidly following fire, or are transient. Secondly, net N cycling rates are often calculated as the difference in pool size between two time points. When measured in the field, repeated sampling of the same soil core would control for spatial heterogeneity in starting conditions, but would



likely distort estimates of N dynamics because soil disturbance can increase rates of C mineralization and microbial respiration. Instead, to avoid disturbance associated with repeated sampling of the same core, nutrients are assumed to be distributed homogeneously in a small sampling area. Thus, it is assumed that cores collected in close proximity to each other are comparable, and can be considered

replicates. However, nutrient pool sizes can vary by orders of magnitude within a meter (Jackson and Caldwell, 1993), and so these assumptions, while practical, are problematic and often invalid. As such, field estimates of net cycling rates calculated from the difference between two nearby cores may be influenced by the idiosyncrasies of N spatial heterogeneity, and may not accurately represent local or larger-scale N dynamics. Without using expensive tracers, field-based sampling protocols to estimate

net nutrient cycling remain imperfect and researchers must collect extensive soil replicates to overcome the issues associated with environmental heterogeneity.

In this study, our broad aim was to evaluate alternative mechanisms driving post-fire changes in N availability while addressing the above mentioned methodological and analytical challenges to estimating net cycling rates. We measured soil N status every week for nine weeks during the 2014

growing season in five longleaf pine savannas sites in North Carolina. Our study is the first that we know of to provide high-resolution temporal (i.e. weekly) data on the effects of prescribed fire on soil N dynamics in longleaf pine savannas. We then used a Bayesian hierarchical linear model to account for heterogeneous *in situ* N availability. The goals of our study were (1) to evaluate the short-term effects of fire on soil inorganic N availability and (2) evaluate whether changes in N pool sizes following fire

could be attributed to changes in net microbial cycling rates or ash deposition.

## 2 Methods

### 2.1 Study Site and Fire Characteristics

Our study was carried out in a longleaf pine savanna on Fort Bragg Military Reservation (35.1391°N 78.9991°W) near Fayetteville, NC, USA. This area is characterized by deep, sandy and sandy loam soils

from the Candor and Blaney series, which lack an O horizon. Mean monthly temperature ranges from 6.9 - 26.0 °C, and mean annual precipitation is 127.5 cm. This area includes numerous





microtopographical gradients represented by numerous low riparian wetlands in an upland matrix. The uplands are well-drained and savanna-like, with an open canopy of longleaf pine (*Pinus palustris*) and an understory dominated by wiregrass (Aristida stricta; Sorrie et al., 2006). Several hardwood species and *Pinus serotina* replace *P. palustris* in the wetlands lining streambeds; in these areas, the soil is often

saturated and the ground covered with *Sphagnum sp.* Separating the uplands from the wetlands, the ecotones have dense, shrubby vegetation dominated by Ericaceous species.

Since the 1980s, prescribed burns have been used as a management tool to maintain the longleaf pine savannas on the reservation; since the mid-1990s, these burns have occurred on 3-year rotations to promote longleaf pine regeneration and maintain habitat for rare and endangered species (Sorrie et al.,

2006). Fort Bragg is composed of burn parcels (hereafter "sites") with independent burn histories. Permanent vegetation sampling transects spanning the topographic gradient have been maintained in 32 sites since 2011; the burn regime in the majority of these sites has been experimentally altered, with a subset of sites being maintained in 3-year burn intervals (Ames et al., 2015). From these sites, we selected three sites scheduled to burn in 2014, and three not schedule to burn in 2014 for use in a

before-after-control-impact experiment. To avoid any artefacts associated with different historical burn characteristics (e.g. historical burns occurring in wetter or dryer years; historical burn intensity and frequency), we limited in the number of burned sites to those with similar burn histories (i.e. all on 3-year burn rotations). However, one site not scheduled to burn until 2016 experienced a wildfire in July 2014, and another site scheduled to burn did not. The site that burned prematurely due to a wildfire was

grouped with other burned sites, despite its shortened fire return interval (one year) relative to the other burned sites (three years). The site that failed to burn was considerably different from the remaining sites (% soil moisture and % soil organic matter were both more than double that of the other sites), and as such was dropped from further analyses. Thus, we were left with 5 study sites: B1, B2, B3 experienced burns in 2014; C1 and C2 were control sites that remained unburned. Our study sites,

renamed here for clarity, correspond to sites 3, 11 (wildfire site), and 15b (B1-3), and 9 and 32 (C1 and C2) described in Ames et al. (2015). At each site, we established a sampling area approximately 5 m upslope of the ecotone. This topographic location was chosen to minimize the effects of extremely well-drained, hydrologically disconnected (as found in the uplands) or saturated, anoxic (as found in the



wetlands) soils on microbial processing. Aboveground vegetation cover was recorded for each species in each site during the growing season of 2014.

Burns occurred in treatment sites B1, B2 (wildfire), B3 on 4 July, 9 July, and 7 July 2014, respectively (Julian days 185, 190, and 188). Using metal tags marked with Tempilaq temperature-sensitive paint

(Air Liquide America 296 Corporation, South Plainfield, NJ, USA), we collected data on aboveground fire temperature at B1 (6 tags) and B3 (8 tags). We did not collect fire temperature data at B2 because it was not initially scheduled to burn.

### 2.2 Soil Analyses

From May 30 through July 25, 2014, soil cores were collected weekly (nine weeks) from each site for

pool size measurements (Figure 1). We collected pool size cores for a minimum of three weeks after a prescribed burn. As such, our site-specific sampling allowed us to collect data before and after burns and detect any immediate changes in N concentration in response to the burn. Each week, three cores (each 5 cm diameter x 12 cm deep and adjacent to each other) were randomly collected from each site, for a total of 27 cores collected for pool size measurements at each site over the nine-week sampling

period. (In burned sites, the number of cores collected prior to versus following prescribed fires differed between sites depending on when the burns occurred. In B1, N=15 cores were collected prior to burns and N=12 cores were collected following prescribed burns. In B2 and B3, N=18 cores were collected prior to burns and N=9 cores were collected following prescribed burns.) These cores were used to compare the pool sizes of nitrate ($NO_3^-$) and $NH_4^+$ throughout the growing season (Figure 1). After

collection, all cores were stored on ice, immediately transported back to the laboratory, and kept at 4 °C until they were analysed for inorganic N, soil moisture, soil organic matter, pH, and $\delta^{15}N$. All cores were homogenized by passing through a 2 mm sieve. Frequent fires in this ecosystem consume aboveground vegetation and litter, preventing the development of an O horizon in these soils (Boring et al., 2004). As such, sieving removed coarse root fragments, rather than partially decomposed organic

matter. Subsamples (~10 g) from each core were extracted within 48 hours of collection with 2 M KCl for inorganic N concentrations. The samples were shaken for 30 minutes, centrifuged, and the extract was then filtered out and stored frozen until analysis on a Lachat QuikChem 8500. Additional soil



subsamples were oven dried for gravimetric soil moisture analysis, and combusted at 450 °C to measure organic matter content. Finally, we measured soil pH in 2:1 dH$_2$O:soil ratios with a bench top pH probe. To measure net cycling rates, we also installed three PVC collars (5 cm internal diameter x 12 cm deep) in each site every week (N=27 cycling rate cores collected at each site over nine-week sampling period;

the number of cores collected prior to versus following the prescribed burns differed between sites depending on when burns occurred, as above). These collars were installed adjacent to the soil cores collected for pool size measures (Figure 1), and were incubated *in situ* for one week, after which time they were collected and analysed as above for inorganic N pool sizes, soil moisture, soil organic matter, and pH. After allowing for error in the initial pool size of the incubated soil core (see below for model

details), net nitrogen cycling rates were calculated based on the difference in extractable NO$_3^-$ and NH$_4^+$ in the incubated cores and the un-incubated cores. That is, while traditional methods assume that the N pool sizes in the initial and incubated soil cores are equivalent, our analyses calculated net cycling rates allowing for differences in initial conditions between the two cores.

## 2.2 δ$^{15}$N Analysis and Mixing Models

To assess plant-derived ash inputs to soil N pools after fire, we analysed soils from a subsample of time points for δ$^{15}$N. Plants generally discriminate against $^{15}$N in favour of $^{14}$N uptake (Craine et al., 2015;Hogberg, 1997), and, as such, vegetation tends to be depleted in $^{15}$N relative to soil. If appreciable plant-derived N was deposited on burned sites as ash, we expected to see a decrease in δ$^{15}$N while observing an increase in N pool size. Although ash is deposited on surficial soils, heavy rains occur

frequently during June-August in this region of North Carolina (11.5, 14.8, and 11.5 cm, respectively; May 25, 2016), and the well-drained, sandy soils could leach nutrients through the soil profile. Moreover, because plants begin resprouting within days of a fire (personal observation), we did not want to discount the potential for nutrients to be redistributed by roots. As a consequence of our uncertainty regarding the vertical distribution of deposited $^{15}$N, we subsampled the full soil cores (0-12

cm) for $^{15}$N analyses. The enrichment of the sample in $^{15}$N is reported on a per mille basis (‰) and was calculated as follows:

$$\delta^{15}\text{N (‰)} = \frac{R_{sample} - R_{standard}}{R_{standard}} \times 1000 \qquad (1)$$





where $R_{sample}$ is the ratio of $^{15}N:^{14}N$ in the sample, and $R_{standard} = 0.0036$, the ratio in atmospheric $N_2$.

We subsampled cores collected at each site for pool size estimates from the two sampling weeks pre-burn and the two sampling weeks post-burn, for a total of four consecutive sampling weeks surrounding each burn (for B1-3, N=6 unburned and N=6 burned samples; for C1 and C2, N=12 samples). For
unburned sites, we subsampled $\delta^{15}N$ for four weeks corresponding to the same consecutive weeks surrounding the burn dates in burned sites (hereafter *burn season*). For example, site B1 burned on July 3, 2014 corresponding to our sixth sampling week. For this site, and for the unburned site C1, we therefore measured $\delta^{15}N$ from soil subsamples in the 4th-7th sampling weeks. Because our sites were not truly paired, we chose time points for $\delta^{15}N$ analyses in the unburned sites based on the burn dates of the
closest burn site. In this way, we allowed ourselves to detect any ash drifting between sites. Soils were oven dried at 40°C until a constant weight, then ground finely, encapsulated in tin capsules, and combusted on a Carlo-Erba Elemental Analyzer coupled to a mass spectrometer at the Duke Environment Isotope Laboratory.

We used two end-member mixing models to estimate the mass of ash-N deposited onto the system. We
used $\delta^{15}N$ and N concentrations from pre-burn soil, and published $\delta^{15}N$ signatures of ash (-0.81 δ15N; Huber et al., 2013) as the end-members, and post-burn soil $\delta^{15}N$ and N concentrations as the resulting mixture from the two end-members. We solved for the mass of ash-N needed to be deposited in order to account for the observed shift in soil $\delta^{15}N$ signature.

To assess whether ash inputs could be detected in post-burn soil cores using the natural abundance of
$^{15}N$, we also calculated the mass N needed to be deposited to observe a shift in soil isotopic signature of the minimum external precision (0.1‰ $\delta^{15}N$ at 1 standard deviation).

### 2.3 Model Development and Statistical Analyses

### 2.3.1 Pool Sizes

To assess the spatial distribution of nutrient ($NO_3^-$ and $NH_4^+$) and SOM availability, we calculated the
coefficient of variation (CV) for each site prior to prescribed burns. To understand how fire and soil variables affect N pool sizes, we fitted a Bayesian hierarchical linear model; this is akin to a multiple regression that also allows for variability in the relationship between true soil N pool sizes (μ) and



measured pool sizes (y) which might occur, for example, though analytical error. Any effect of soil environmental conditions on N pool sizes would occur on $\mu$, not $y$. Each core was modelled independently, and we accounted for site blocking effects by including random intercepts for each site. For core i = 1…. n at site j = 1…5, observed N pool size ($NH_4^+$ or $NO_3^-$) was modelled as a function of random site effects, percent soil moisture (SM), percent soil organic matter (SOM), soil pH, and the number of days since the previous burn, days since fire (DSF) as follows

$$y_{0_{i,j}} \sim N(\mu_{0_{i,j}}, \sigma^2) \tag{2a}$$

$$\mu_{0_{i,j}} = \beta_{0_{i,j}} + \beta_{1_i}SM_i + \beta_{2_i}SOM_i + \beta_{3_i}pH_i + \beta_{4_i}DSF^{-1}_i \tag{2b}$$

For $NO_3^-$, we added an additional predictive parameter, $\beta_{5_i}NH_{4\ i}^+$ to allow for $NO_3^-$ concentrations to additionally vary with nitrification substrate ($NH_4^+$) availability. We expected the effects of a burn to diminish with time, so we transformed DSF to $DSF^{-1}$, so that as DSF increased, $DSF^{-1} \rightarrow 0$. Full models included all main effects and no interactions.

### 2.3.2 Cycling Rates

We built a hierarchical state-space model within a Bayesian framework to understand how fire and soil variables affected N cycling rates. As in the models of N pool sizes above, our cycling rate models allowed for variation in the relationship between true ($\mu$) and measured (z) cycling rates, modeled below as $\tau$. In addition, we also allowed for error associated with the assumption that the N concentrations in initial cores ($y_0$; as from equation 2a) were equal to the initial concentrations of the incubating cores ($y_1$). By including these errors into our model, we essentially relaxed the assumption that paired cores (un-incubated and incubated cores) were true replicates and had equal initial N concentration and edaphic conditions (SOM, pH, SM). We removed four core pairs (of 135) that exhibited $NH_4^+$ or $NO_3^-$ concentrations below the detection limit in the initial concentration. For core i = 1…n at each site j = 1…5, cycling rate was modelled as

$$z_{i,j} \sim N(\mu_{i,j}, \tau) \tag{3a}$$

$$\mu_{i,j} = y_{1_{i,j}} - y_{0_{i,j}} = \beta_{1_i}SM_i + \beta_{2_i}SOM_i + \beta_{3_i}pH_i + \beta_{4_i}DSF^{-1}_i + \beta_{5_i}D_i + \beta_{6_i}y_{0_{i,j}} \tag{3b}$$

$$\tau = \sigma^{-2} \tag{3c}$$





$\sigma \sim \text{unif}(0,100)$ (3d)

To model cycling rates, we included incubation length (D, in days) and initial $NH_4^+$ or $NO_3^-$ concentration as additional predictors. Again, full models included all main effects and no interactions. Prior to all analyses, we removed two cores from B1 (burned) and two from C2 that had pool size

values below the analytical detection limit (four of 135 cores). All models were built with the rjags package (version 3.15) in R version 3.2.1 (R Development Core Team, 2011). All predictors were modelled with normally distributed, uninformative priors. All values are reported with 95% credible intervals (CI).

### 3 Results

### 3.2 Site Conditions and Fire Characteristics

Although plant community composition varied, *Gaylusacia frondosa*, *Clethra alnifolia*, and *Arudinaria tecta* were dominant in all of our study sites. Study sites were dry, low in organic matter, and acidic (Figure 2). Prescribed burns in all three sites thoroughly consumed all or most of the aboveground biomass. Aboveground understory vegetation in B1 and B3 was completely consumed. In B2 some

scorched leaves remained on the woody vegetation, but the herbaceous understory species were completely consumed. Fire temperatures were similar between B1 and B3: average maximum fire temperature at B1 was 612 °C ±18 and at B3 was 635 °C ±18. Fire temperature was not measured in B2, but is generally affected by fuel load and moisture (Ellair and Platt, 2012).

### 3.3 Pool Sizes

To assess the fine-scale spatial variability in $NH_4^+$ and $NO_3^-$ concentrations, we compared the CV of $NO_3^-$ and $NH_4^+$ pool sizes within each site prior to prescribed burns. Both $NO_3^-$ and $NH_4^+$ pool sizes were highly spatially heterogeneous, despite similar mean concentrations across sites prior to prescribed burns (Figures 2-3). Spatial variability in $NO_3^-$ pool size was high within each site, but each site exhibited similar variation in $NO_3^-$ pool size. CV-$NO_3$ ranged from 42.3 in B1 to 57.6 in C2. $NH_4^+$ pool

sizes were also highly variable across sites, and there was a considerable range of the spatial heterogeneity across sites. CV-$NH_4^+$ ranged from 57.0 in B2 to 114.2 in C2.




In the first week of sampling, initial pool sizes of inorganic N were similar between sites. Over the pre-burn season, sites had greater $NO_3^-$ than $NH_4^+$ availability (3.06 ±0.16 µg $NO_3^-$ per gram of dry soil [gds$^{-1}$] and 0.86 ±0.07 µg $NH_4^+$ gds$^{-1}$). However, the ratio of $NH_4^+$ : $NO_3^-$ increased following prescribed burns and sites B2 and B3 both experienced a shift in the dominant inorganic N form to

$NH_4^+$ immediately after a burn.

There were observable increases in $NH_4^+$ pool sizes immediately after a burn relative to the same time points in unburned control sites and time points in burned sites immediately prior to the burn (Figure 6). Three days post-burn in Site B1, $NH_4^+$ had increased from 0.83 ±0.15 µg $NH_4^+$ gds$^{-1}$ in the pre-burn season to 6.10 ±1.08 µg $NH_4^+$ gds$^{-1}$. One day post-burn in Site B2, $NH_4^+$ had increased from 0.91 ±0.12

µg $NH_4^+$ gds$^{-1}$ in the pre-burn season to 7.79 ±2.09 µg $NH_4^+$ gds$^{-1}$. The pattern of $NH_4^+$ pool size change in Site B3 was qualitatively different than changes observed in Sites B1 and B2. In Site B3, there was an approximate exponential increase in $NH_4^+$ pool size that plateaued, but did not diminish, by the end of our sampling, more than 3 weeks post-burn. Three days following a burn in Site B3, $NH_4^+$ pool size had increased from 0.84 ±0.16 µg $NH_4^+$ gds$^{-1}$ in the pre-burn season to 21.9 ±6.31 µg $NH_4^+$ gds$^{-1}$.

Pool sizes of $NH_4^+$ were most strongly correlated with days since fire (DSF; $\beta_{DSF}$ = 12.50, 95% credible interval (CI) = 5.34-19.66; Figure 6) and pH ($\beta_{pH}$ = 6.16, 95% CI = 1.79-10.46; Figure 6). Because we fit our pool size models to the inverse of DSF (i.e. DSF$^{-1}$; see *Methods*), the positive correlation between $NH_4^+$ and DSF indicates decreasing pool sizes as time since fire lengthens. Pool sizes of $NH_4^+$ were larger for less acidic soils and in recently burned soils. Pool sizes of $NH_4^+$ were slightly greater in

soils with more organic matter ($\beta_{SOM}$ = 0.2, 95% CI = 0.01-0.30), but did not vary with soil moisture ($\beta_{SM}$ = -0.11, 95% CI = -0.30-0.07; Figure 6).

In contrast to observed pulses of $NH_4^+$ availability following fire, we did not find strong fire-associated increases in $NO_3^-$ pool size (Figure 5). $NO_3^-$ availability ranged from 0.1 to 8.56 µg $NO_3^-$ gds$^{-1}$, and on average was 2.85 ±0.15 µg $NO_3^-$ gds$^{-1}$. DSF was not related to $NO_3^-$ pool sizes ($\beta_{DSF}$ = 0.44, 95% CI =

-1.22-2.10; Figure 6). Pool sizes of $NO_3^-$ varied with soil organic matter content ($\beta_{SOM}$ = 0.03, 95% CI = 0.01-0.07), $NH_4^+$ pool size ($\beta_{NH4+}$ = 0.06, 95% CI = 0.02-0.10), and pH ($\beta_{pH}$ = 1.46, 95% CI = 0.39-2.49; Figure 6).



We also examined whether there was a long-term legacy of fire detectable in sites that had not experienced prescribed burns in that growing season (i.e. across pre-burn time points in burned sites, and all time points from unburned sites). In these areas that had not experienced recent fires, DSF had no effect on $NH_4^+$ ($\beta_{DSF}$ = 0.70, 95% CI = -60.54-61.46) or $NO_3^-$ availability ($\beta_{DSF}$ = 1.79, 95% CI =

-60.31-63.75). Soil moisture ($\beta_{SM}$ = 1.55, 95% CI = 0.65-2.45) and, to lesser extents, $NH_4^+$ availability ($\beta_{NH4+}$ = 0.19, 95% CI = 0.45-0.72) and soil organic matter ($\beta_{SOM}$ = 0.03, 95% CI = 0.01-0.07) were positively correlated with $NO_3^-$ pool sizes.

### 3.3 Cycling Rates

Net N cycling rates were generally low and temporally heterogeneous across the growing season

(Figure 7). They varied between net production and net consumption between sampling points (Figure 7). Across the whole growing season, net mineralization in unburned sites was 0.19 (±0.16) μg gds$^{-1}$ day$^{-1}$; in burned sites, it was 0.26 (±0.05) μg gds$^{-1}$ day$^{-1}$. Net mineralization rates were appreciably more variable the week following a burn, but patterns between sites were inconsistent (Figure 7). For the first week following burns in sites B1-3, net mineralization rates were -1.72 (±0.32), 0.31 (±0.10), and 2.54

(±0.46) μg gds$^{-1}$ day$^{-1}$ respectively. Despite the change in net mineralization pattern following a burn, there was no consistent effect of DSF on net mineralization rate (Figure 8; $\beta_{DSF}$ = 0.34, 95% CI = -0.50-1.19). Of the measured edaphic variables, net mineralization rates were correlated only slightly with soil moisture (Figure 8; $\beta_{SM}$ = 0.31, 95% CI = 0.14-0.48).

Net nitrification rates were temporally heterogeneous throughout the full growing season, but were not

appreciably more variable immediately following burns. Net nitrification rates were very low across the growing season in burned (-0.08 ±0.05 μg gds$^{-1}$ day$^{-1}$) and unburned sites (-0.04 ±0.04 μg gds$^{-1}$ day$^{-1}$; Figure 7). Measured edaphic parameters were poorly correlated with observed net nitrification rates, although there was a slight positive relationship between net nitrification rates and soil moisture (Figure 8; $\beta_{SM}$ = 0.07, 95% CI = 0.03-0.11).



### 3.4 Total Soil $\delta^{15}$N and Ash Deposition

Soil N concentration was relatively stable throughout the burn season, and was similar between burned (0.35%N ±0.09) and unburned sites (0.38%N ±0.07; Table 1). Mean total soil N varied between sites (Table 1; Figure 9). On average, B2 had the lowest total soil N content (0.18%N ±0.02), and total soil N

ranged from 0.08% at C1, to 0.98% at C2. Across all burned and unburned time points, confidence intervals overlapped between burned and unburned conditions at each site, indicating no persistent change in total N over the full growing season (Table 1).

Across the full growing season, soil $\delta^{15}$N in burned sites was 2.76‰ (±0.36) and in unburned sites was 2.00‰ (±0.36). The response of $\delta^{15}$N to burning varied between sites. Soil $\delta^{15}$N in unburned Sites C1

and C2 was on average 1.22‰ (±0.52) and 1.79‰ (±0.36; Table 1), respectively. In burned sites, there were shifts in $\delta^{15}$N, although the direction of shift varied between sites. Soils in B1 were depleted in $^{15}$N after a prescribed burn relative to before the burn; $\delta^{15}$N shifted from 3.98‰ (±0.64) to 3.22‰ (±0.50). In Site B2 the soil $\delta^{15}$N decreased from 2.67‰ (±0.57) before the prescribed burn to 2.45‰ (±0.24) after the burn. Finally, there was a slight enrichment in soil $^{15}$N in Site B3 following fire; $\delta^{15}$N shifted

from 1.37‰ (±0.36; Table 1) to 2.59‰ (±0.93; Table 1). The 95% CIs surrounding the mean $\delta^{15}$N (and %N) overlapped for all burned sites, indicating the soil $\delta^{15}$N at each site was statistically indistinguishable pre- and post-burn.

We used pre-burn soil $\delta^{15}$N isotopic signature in mixing models to calculate the mass of ash-N needed to be deposited on our sites to achieve both the minimum and the empirically measured shift in soil

$\delta^{15}$N. To achieve a shift in soil $\delta^{15}$N of the minimum external precision, sites B1, B2, and B3 would need 11, 5, and 20 g N m$^{-2}$ ash-N, respectively, deposited following fire. To achieve the measured shift in soil $\delta^{15}$N, sites B1 and B2 would need 100 and 11 g N m$^{-2}$ added through ash deposition; B3 would need 175 g N m$^{-2}$ to be removed from fire (Table 1). We also calculated the same values using fresh leaf $\delta^{15}$N from leaves collected from our sample site (-2.9‰ ±0.1; J. Wright, unpublished data), rather than

published ash $\delta^{15}$N values (Supplementary Table S1).



## 4 Discussion

In this study, we collected weekly measurements of soil inorganic N availability to document short-lived changes in N dynamics following fire and throughout the growing season of a pyrogenic forest in the southeastern US. As far as we know, this is the first study to pair estimates of N pool sizes and

cycling rates at high temporal resolution in a longleaf pine savanna. Prior to prescribed burns, there was high variability in N availability, particularly for $NH_4^+$ pool sizes. This heterogeneity reinforces the need for a methodological approach that considers initial edaphic conditions when carrying out *in situ* experiments on paired soil cores. To address this, we relaxed the assumption that initial and incubating cores were true edaphic replicates; we used a Bayesian statistical framework to allow for variability in

the relationship between true versus measured inorganic N concentrations in our soils.

Immediately following prescribed burns, we found sharp increases in $NH_4^+$ pool sizes in all of our study sites. However, the magnitude and duration of this increase varied between sites. Unlike studies in southeastern US pine savannas with monthly or less-frequent temporal sampling protocols, our weekly sampling allowed us to capture highly ephemeral changes in soil inorganic N pools. Furthermore, we

found no changes in cycling rates and no evidence that ash deposition could account for the large increases in N availability following fire. Instead, we propose that an ephemeral dampening of plant uptake could contribute to the observed increases in inorganic N following fire.

### 4.1 Changes in N dynamics across the growing season

Throughout the growing season, inorganic N availability and net cycling rates were low, as is common

in longleaf pine savannas (Binkley et al., 1992). In unburned conditions over the growing season, there was greater $NO_3^-$ availability than $NH_4^+$. This pattern is consistent with previous work, which documented relatively high $NH_4^+$ availability in the winter, followed by decreasing $NH_4^+$ availability throughout the growing season (Christensen, 1977). Net nitrification rates were low across the growing season, may have been inhibited by the low soil pH. Net mineralization in our study was higher than

measured over the summer months in previous studies (Wilson et al., 1999), so rather than low mineralization rates, our low soil $NH_4^+ : NO_3^-$ ratios may be a result of preferential plant or microbial uptake of $NH_4^+$ over $NO_3^-$.



We observed sharp increases in soil inorganic $NH_4^+$, but not $NO_3^-$, immediately following fire across three longleaf pine savanna sites in North Carolina (Figure 5). Although a global meta-analysis found that post-fire soil $NO_3^-$ concentrations peak ten months after $NH_4^+$ concentrations (Wan et al., 2001), studies in southeastern US ecosystems found no change in soil $NO_3^-$ up to 30 days (pine savanna;

Boring et al., 2004), 320 days (pine forest; Lavoie et al., 2010) and 500 days (shrubland; Schafer and Mack, 2010) following fire. Across the full growing season, we measured $NH_4^+$ pool sizes of burned sites that were nearly 5x that of unburned sites. The direction of the effect of fire was consistent across our study sites, however the magnitude of increase was highly site-specific. Within a site, increases in $NH_4^+$ availability immediately following fire ranged from 5x to more than 25x the pre-burn levels. This

$NH_4^+$ pulse was short-lived, and only in B3 was the increased $NH_4^+$ pool size sustained for longer than one week. As a consequence, a decreased sampling frequency would not have detected the ephemeral changes in soil $NH_4^+$ pool size in B1 and B2.

It remains unclear why, however, sites experience such variability in the magnitude of $NH_4^+$ response following fire. Although we cannot rule out the possibility that our high-intensity sampling influenced

nitrogen cycling and pool sizes, we saw no evidence of increasing inorganic N availability, or increasing variability in N availability, in our control sites, which experienced the same levels of sampling disturbance without fire. However, differences in post-fire vegetation regrowth in sites B1, B2, and B3 (C. Ficken, unpublished data), may suggest an important role of plant uptake in regulating soil N concentration. While B1 and B2 exhibited rapid vegetation resprouting following fire, regrowth

in B3 was patchy. Moreover, in 2012, the last year all sites were sampled prior to the 2014 burns, B3 had the smallest standing biomass stocks of all three burned sites (J. Wright unpublished data). If plant N uptake remained low following fire in B3, this might explain the persistent increase in N availability in this site. However, biomass stocks after three months of regrowth in 2014 were also substantially different between B1 and B2, despite these sites exhibiting similar patterns of $NH_4^+$ availability over

time. In unburned years, Mitchell et al. (1999) found that annual net primary productivity (ANPP) in a longleaf pine savanna was positively correlated with local moisture availability and biomass estimates in this heterogeneous system are highly dependent on local woody versus herbaceous cover, as well as annual variability in environmental conditions. Factors controlling unburned ANPP may differ from



those controlling biomass regeneration, and given the spatiotemporal heterogeneity of this system, teasing apart these drivers may require a large-scale manipulative experiment.

**4.2 Assessing Mechanisms of N Pulses Following Fire**

A nitrogen pulse may occur following fire through (1) an increase in microbial mineralization, (2) ash
inputs, or (3) a decrease in plant uptake. Fire may stimulate microbial turnover of organic matter if additions of C or N from ash deposition or root exudation (southern shrubland; Schafer and Mack, 2010) enhance microbial activity. Wilson et al (2002) found significant increases in microbial biomass following fire in a longleaf pine savanna. Although we did not directly measure microbial biomass, we found no changes in net microbial mineralization associated with the observed increase in pool size,
although cycling rates were increasingly variable following fire in burned sites (but not in unburned sites). Indeed, the increase in mass of $NH_4^+$ following fire was much greater than the mass of $NH_4^+$ produced on a daily basis by net microbial mineralization, reinforcing the conclusion that changes in microbial cycling rates could not account for the observed increase in pool sizes.

We also found no indication that the newly available $NH_4^+$ substrate led to a delayed increase in net
nitrification rate. This, along with high soil C:N (unburned sites- 47:1; burned sites- 53:1) relative to other longleaf pine soils (Lucash et al., 2007), might suggest that autotrophic nitrifying microbes are competitively inferior to heterotrophic microbes under post-fire conditions in our study sites. Alternatively, an unmeasured increase in gross nitrification might have allowed for a commensurate increase in microbial immobilization of $NO_3^-$ following fire. In general, however, the lack of change in
microbial N cycling rates suggests that changes in microbial activity fail to account for the observed increases in $NH_4^+$ availability following fire.

Direct additions of N into the soil from ash may provide an alternative mechanism for the observed increase in mineral N availability. To test this, we examined changes in both total N and $\delta^{15}N$ immediately before and after burns. N from ash additions is primarily organic (Christensen, 1977;Huber
et al., 2013;Raison, 1979), and is thought to increase N pools by stimulating microbial activity. We found no change in total nitrogen (i.e. %N) or soil organic matter before and after prescribed burns.



These results support the findings of a global meta-analysis of fires in forested systems, which found no effect of fire on total N (Wan et al., 2001).

We used the natural abundance of $^{15}$N as an isotopic tracer of ash additions. However, fractionation during volatization preferentially releases $^{14}$N, resulting in ash material that is enriched in $^{15}$N relative to
fresh plant matter and an increase in $\delta^{15}$N signature in ash relative to fresh plant material (Saito et al., 2007;Stephan et al., 2015). A study in a subalpine grassland reported foliar $\delta^{15}$N values (-2.9‰) and N concentrations in ash (11.63 ±0.80 mg N g-1 ash; Huber et al., 2013) comparable to foliar $\delta^{15}$N values (-2.9‰; ±0.1; J. Wright, unpublished data) and ash-N concentrations of our system (8.75 ±0.90 mg N g-1 ash; Christensen, 1977). Huber et al. (2013) also reported ash $\delta^{15}$N values in ash of -0.81‰, which
we used in mixing models. Using this isotopic signature of ash, we found that 5 to 20 g ash-N m$^{-2}$ would need to be deposited in our burned sites in order to observe a detectable shift in soil isotopic signature. These values are greater than the mass of ash-N deposition reported in a longleaf pine system (1.15 g m-2; Christensen, 1977), suggesting that this method may not be ideal for detecting ash inputs in systems with low aboveground vegetation stocks. Nevertheless, we estimated that 100, 11, and 175 g
ash-N m$^{-2}$ would need to be deposited on sites B1-3 to account for our measured shifts in soil $\delta^{15}$N. These deposition levels are highly unlikely to have occurred at our sites, since they would require substantial aboveground vegetation accumulation and our system is burned every three years. However, it is unclear how quickly surface inputs can be expected to distribute throughout the soil profile. In the sandy soils of our system, frequent heavy summer rains or active root growth may quickly redistribute
surface inputs. While one study of longleaf pine savannas found that changes (losses) in total soil N following fire were concentrated at the soil surface (Binkley et al., 1992), another study of subalpine woodlands and grasslands detected no changes in total N in surface soils following burning (Huber et al., 2013). Our work is also in agreement with that of (Christensen, 1977), who found significant differences in $\delta^{15}$N with depth, but no change following fire. Given the uncertainties surrounding the
redistribution of surface inputs down the soil profile, we cannot conclusively rule out the potential to surface additions to contribute to the observed $NH_4^+$ pulse. Nevertheless, considering the unrealistic mass of ash-N needed to be deposited onto surface soils to account for our measured shifts in $\delta^{15}$N, we



conclude that ash inputs are unlikely to fully account for the increase in measured soil inorganic N availability.

Finally, changes in plant and microbial immobilization could cause an increase in soil inorganic pool sizes. Prescribed fires in longleaf pine savannas are low intensity, and sharp declines in soil temperature

with depth, particularly in dry soils, are unlikely to substantially damage the soil microbial community below 5 cm (Hartford and Frandsen, 1992). In fact, previous work in longleaf pine savannas has documented increases in microbial biomass following fire (Wilson et al 2002). In contrast to the microbial community, prescribed burns in our study system generally topkill a majority of the aboveground herbaceous and woody biomass with stem diameters less than 10 cm (Just et al., 2015). If

fire damage temporarily halted or slowed the plant uptake of inorganic N, we would expect to see an accumulation of soil N if microbial immobilization did not increase sufficiently to deplete the pool. However, N accumulating in excess of demand can only partly explain observed increases in inorganic N availability, since the pulse of N we detected following fire was many times greater than what was produced by net mineralization and net nitrification. Nevertheless, a change in plant sink strength may

have contributed to post-fire $NH_4^+$ pulse.

Previous work found no evidence that plant species in an African savanna re-translocated nutrients from root biomass to resprouting shoot biomass following a fire (Vijver et al., 1999), indicating that soil pools can be an important source of N for regenerating biomass. Indeed, the biomass of resprouting vegetation following fire has been shown to be highly enriched in $^{15}N$ relative to pre-burn biomass

(Huber et al., 2013;Schafer and Mack, 2014), an indication that the source of N in resprouting biomass is also enriched (Evans, 2001). Root biomass is an important component of short-term N retention in grassland ecosystems (De Vries and Bardgett, 2016). In fire-prone systems, fire-tolerant plants could play an important role in preventing N leaching losses if they are able to resume N uptake quickly following fire.

We propose that plant-demand for inorganic N may have a strong influence on soil N pool sizes in this system, and a temporary decrease in this demand may have contributed to the observed post-fire $NH_4^+$ pulse. Plant preference for $NH_4^+$ would explain the relatively large pool sizes of $NO_3^-$ relative to $NH_4^+$ during the growing season, and this pattern is consistent with previous seasonal trends in a longleaf pine



savannas (Christensen, 1977). Furthermore, an increase in soil $NH_4^+$ pool size after fire without a stable increase in net microbial mineralization rates could occur if there is a decrease in plant uptake. Similarly, a muted effect of fire on $NO_3^-$ pool size may occur if plants have diminished uptake of this inorganic N form, and plant uptake exerts a relatively weaker control on soil $NO_3^-$ pools. Plant control

on ecosystem N status has been well documented in northeastern US hardwood forests, where a defoliation event resulted in substantial N losses from the ecosystem (Aber et al., 2002;Likens et al., 1969).

If post-fire patterns in N availability were related to plant uptake, we would expect differences in the magnitude and duration of soil N change following fire to be related to plant N-demand and regrowth

following fire. In stand-replacing fires in temperate forests, where vegetation is killed, relatively persistent increases in N pools should occur following fire. We similarly would expect smaller and more ephemeral changes in N pools in systems in which plants are only top-killed. Indeed, stand-replacing fires have been shown to result in changes to soil N pools that persist more than one year following the fires (Smithwick et al., 2005;Turner et al., 2007). In contrast, elevated inorganic N

immediately following fires in grassland decreases throughout the growing season (Augustine et al., 2014). Low-intensity fires in grass-dominated glades adjacent to oak-hickory forest sites in a Kentucky study resulted in increases in post-burn soil $NO_3^-$ pool sizes, but no increase in lysimeter-detected $NO_3^-$ leaching losses below 10 cm (Trammell et al., 2004). Although the study did not examine microbial biomass, they found no effect of fire on net N mineralization, suggesting role of plant uptake in patterns

of N loss and retention post-fire. Such instances of plant control of N availability provide an important setting in which to examine the role of nutrient availability– and nutrient pulses in particular– on plant community composition and ecosystem productivity. Differences in the ability of species to capture this ephemeral resource may help explain differences in post-fire resprouting patterns and biomass regeneration following fire.

**4.3 Ecological Implications of Fire-Associated N Pulse**

To put the fire-associated pulse of inorganic N that we observed into context, we compared its mass to N inputs in the longleaf pine savannas ecosystem. Although the pulse of soil N following fire is most




likely a redistribution of N from other ecosystem pools, it is conceptually helpful to understand the magnitude of this pulse relative to other components of the N cycle in longleaf pine savannas. The increase in soil inorganic N following fire (0.98 g N m$^{-2}$ 10 cm$^{-1}$) was approximately 10x the daily total net inorganic N production (i.e. net mineralization + net nitrification; 0.11 g N m$^{-2}$ 10 cm$^{-1}$ day$^{-1}$).

These ephemeral increases in soil inorganic N availability occur during an important ontological stage of plant development as longleaf pine understory species begin resprouting within a few days following fire (CD Ficken, personal observation). The mass of mineral N released following fire was more than twice the mass of N longleaf pine understory species assimilate into their biomass over a full growing season (0.45 g N m$^{-2}$; unpublished data). However, the extent to which plants or microbes have access

to this ephemeral nutrient pulse remains unclear. Previous studies have documented rapid plant uptake of N tracers by intact plants (Aber et al., 2002;Likens et al., 1969), suggesting that plants may have access to this N pulse during regrowth.

**4.4 Conclusions**

We have documented large pulses of mineral N following prescribed burns in a longleaf pine savanna in

North Carolina. Our weekly sampling revealed that while one site experienced a persistent increase in NH$_4^+$ pool size, other sites experienced only very short-lived pulses that would not have been detected with monthly sampling. The marked differences in the duration of the NH$_4^+$ pulse that we observed may explain why previous studies with less-frequent soil sampling showed no change in mineral N following fire. However, the factors that influence the magnitude of the system's response to fire are still

unknown. We propose here a role for plant uptake in regulating post-fire N availability, and encourage future work to explore the relationships between N availability and plant biomass dynamics immediately following fire. Due to the rapid changes in N availability following fire, as well as the fast resprouting of understory species, we recommend that responses of the the local plant community be considered when determining an appropriate sampling regime for biogeochemical responses to

disturbance. In systems in which the plant community responds rapidly, soil samples should be collectedly quickly and frequently to capture post-disturbance plant-nutrient and biogeochemical dynamics.

The pulses documented here were not associated with increases in microbial activity, and, although inconclusive, our data also do not support the hypothesis that the observed inorganic N pulse could be attributed to ash inputs to the system. We propose that a temporary slowing of plant uptake may contribute to the observed mineral N pulse. Although plants begin resprouting a few days after a fire,

the extent to which plants access the newly available $NH_4^+$ remains unclear. Given the magnitude of this pulse relative to other N transformations in the system, and particularly relative to the mass of N estimated to be assimilated into the understory biomass annually, these fire-associated N pulses may be important sources of plant-available N as the vegetation regrows. As such, they may also play an important role in structuring plant recovery from disturbance and shaping community diversity in this

system.

## Author Contributions

CDF and JPW conceived of the project; CDF collected samples, performed laboratory and statistical analyses; CDF and JPW wrote the manuscript.

## Acknowledgements

Funding for this project was provided by grant W9132T-11-2-0008 from the U.S. Army Corps of Engineer Research Development Center, Construction Engineering Research Laboratory (ERDC-CERL) to JPW, and by a Sigma Xi grant to CDF. The authors would like to thank Matt Hohmann with the Army Corps of Engineers for substantial constructive feedback throughout manuscript preparation, Janet Gray from the Fort Bragg Endangered Species Branch for logistical support, and Ben Grunwald

for help with soil extractions. The authors declare no conflicts of interest.

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



| | | Burned Sites | | | Unburned Sites | |
|---|---|---|---|---|---|---|
| | | B1 | B2 | B3 | C1 | C2 |
| $NH_4^+$ | Preburn | 0.83 (0.29) | 0.91 (0.24) | 0.84 (0.31) | 1.25 (0.28) | 0.80 (0.14) |
| ($\mu$g gds$^{-1}$) | Postburn | 2.78 (1.62) | 5.18 (2.21) | 29.68 (6.03) | | |
| $NO_3^-$ | Preburn | 3.63 (0.78) | 2.61 (0.63) | 2.74 (0.63) | 2.34 (0.26) | 2.44 (0.22) |
| ($\mu$g gds$^{-1}$) | Postburn | 4.18 (1.76) | 1.69 (0.65) | 4.66 (1.35) | | |
| $\delta^{15}N$ (‰) | Preburn | 3.98 (0.64) | 2.67 (0.57) | 1.37 (0.36) | 1.22 (0.52) | 1.79 (0.36) |
| | Postburn | 3.22 (0.50) | 2.45 (0.24) | 2.60 (0.93) | | |
| Total N (%) | Preburn | 0.61 (0.18) | 0.19 (0.04) | 0.54 (0.20) | 0.29 (0.10) | 0.39 (0.14) |
| | Postburn | 0.56 (0.12) | 0.18 (0.02) | 0.30 (0.11) | | |
| Ash-N (g N m$^{-2}$) | | 104 | 11 | -175 | | |

**Table 1. Mean soil inorganic N pool sizes reported per gram of dry soil (gds), $\delta^{15}N$ values, and total N for each study site. Values are reported ±95% CI. Sample sizes for inorganic N pool sizes differ between sites depending on when the prescribed burns occurred; see *Methods* for details on sample sizes. Soil %N and $\delta^{15}N$ values were collected on a subset of time points; sample sizes for these variables are N=6 for burned site means, and N=12 for control (unburned) site means. Ash-derived N values are the estimated masses of N needed to be deposited onto each site to result in the observed post-burn $\delta^{15}N$. Note that Christensen (1977) measured 1.15 ±0.49 g N m$^{-2}$ deposited in ash fall in fire in a longleaf pine savannas.**



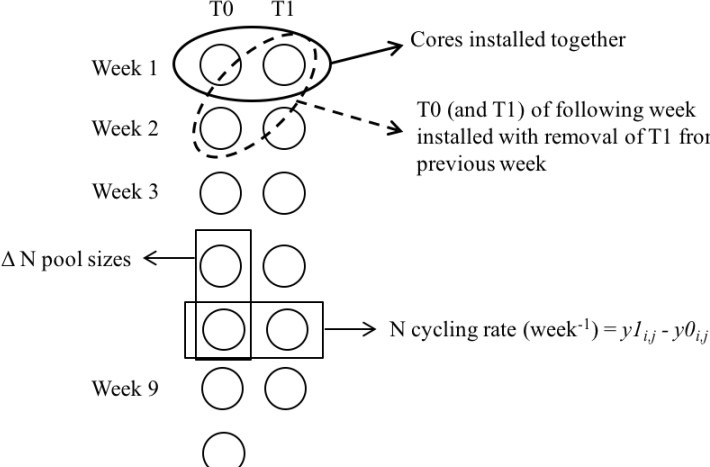

**Figure 1. Schematic illustrating the paired-core sampling design across nine weeks throughout the growing season. Each circle represents the three soil cores and three incubated PVC collars installed every week.**





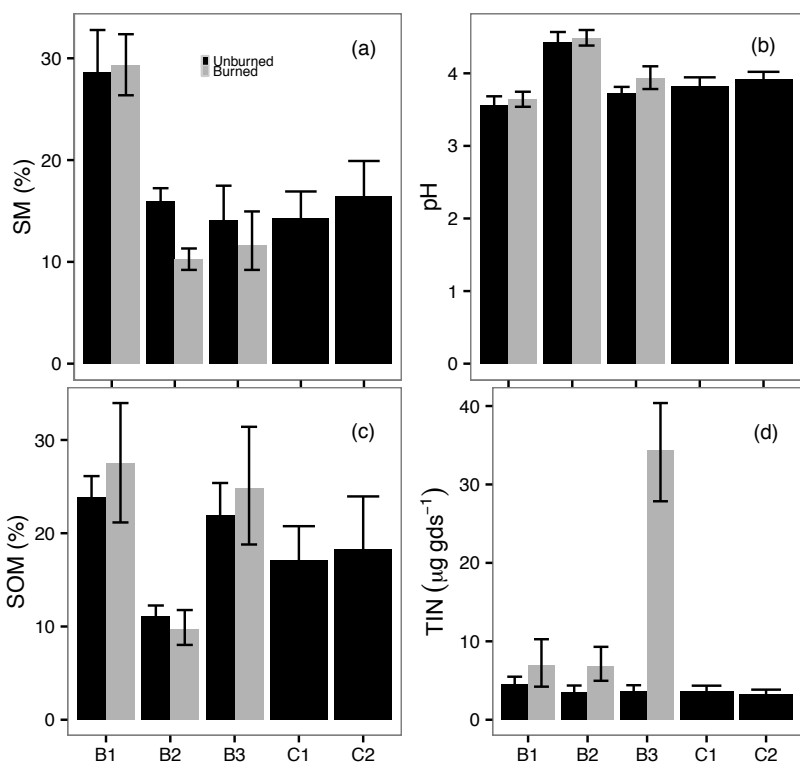

**Figure 2. Mean soil moisture (SM; a), pH (b), soil organic matter (SOM; c) and total inorganic N (TIN, i.e. NH$_4^+$ and NO$_3^-$; d) prior to and following burns at each site. TIN is reported per gram of dry soil (gds). Values are reported ±95% CI.**



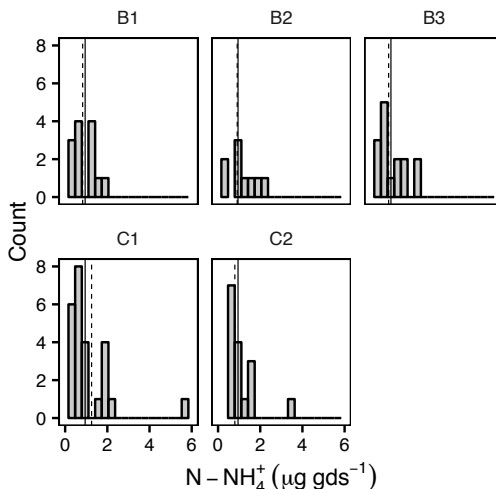

**Figure 3. Histogram of soil $NH_4^+$ concentrations (reported per gram of dry soil, gds) prior to prescribed burns. Solid vertical lines indicate the median concentration across all sites; dashed vertical lines indicate the site-specific median concentration.**

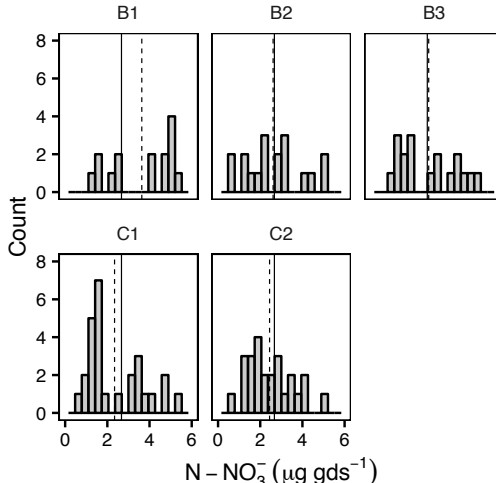

5 **Figure 4. Histogram of soil $NO_3^-$ concentrations (reported per gram of dry soil, gds) prior to prescribed burns. Solid vertical lines indicate the median concentration across all sites; dashed vertical lines indicate the site-specific median concentration.**



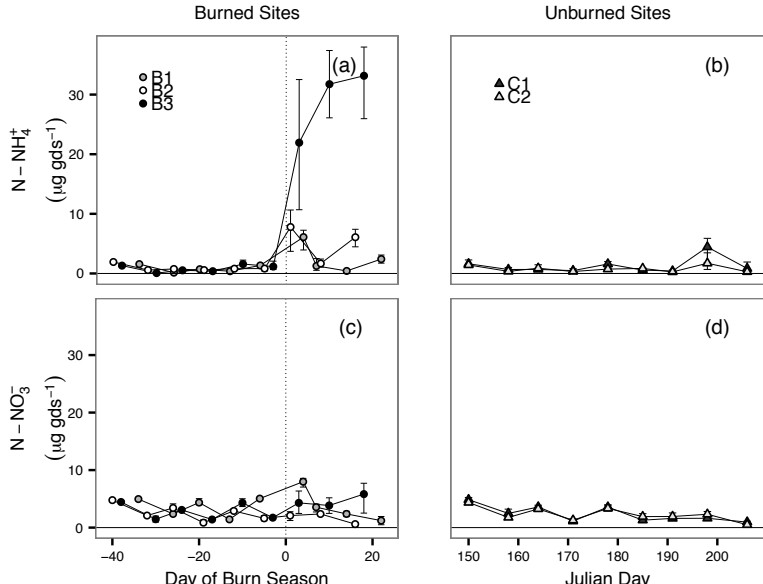

**Figure 5. Changes in pool sizes of (a) $NH_4^+$ in burned sites, (b) $NH_4^+$ in control sites, (c) $NO_3^-$ in burned sites, and (d) $NO_3^-$ in control sites. The x-axis for A and C depicts time (in days) centred on the date of burn; days immediately before the burn are negative x-values while days immediately following a burn are positive; the burn date is at 0 and is demarcated with a vertical dotted line. The x-axis for sites B and D depicts time in Julian Days. Prescribed burns in B1, B2, and B3 occurred on Julian Days 185, 190, and 188, respectively. Pool sizes are reported in in µg N per gram dry soil (gds). Error bars (±95% CI) for individual time points may be obscured by the point.**





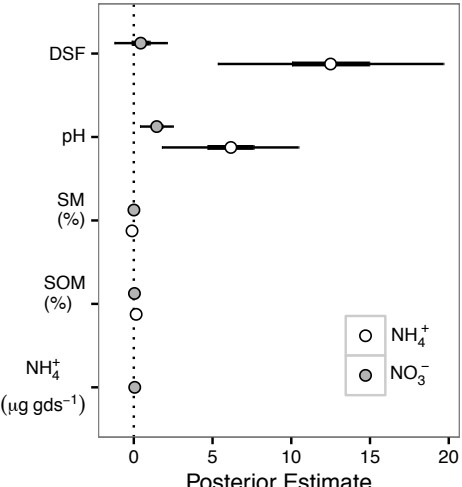

**Figure 6. Mean posterior estimates for parameters predicting pool sizes of NH$_4^+$ and NO$_3^-$. Parameters are days since fire (DSF), soil pH, soil moisture (SM), soil organic matter (SOM), and substrate availability (i.e. NH$_4^+$ reported per gram of dry soil, gds). Thin black lines show 95% CI and thick lines show 50% CI. CI may be obscured by the mean point.**





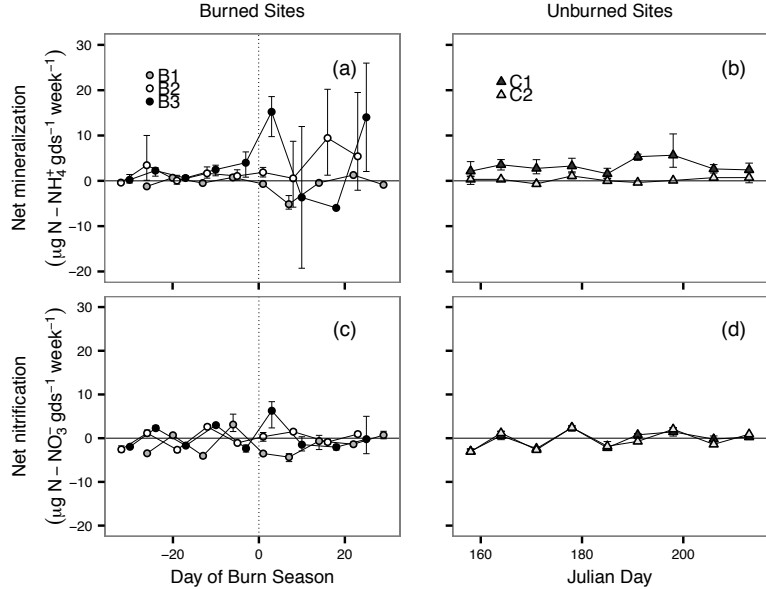

**Figure 7. Changes in (a) net mineralization in burned sites, (b) net mineralization in unburned sites, (c) net nitrification in burned sites and (d) net nitrification in unburned sites. The x-axis for A and C depicts time (in days) centred on the date of burn; days immediately before the burn are negative x-values while days immediately following a burn are positive; the burn date is at 0 and**
5 **is demarcated with a vertical dotted line. The x-axis for sites B and D depicts time in Julian Days. Prescribed burns in B1, B2, and B3 occurred on Julian Days 185, 190, and 188, respectively. Cycling rates are reported in μg N per gram dry soil (gds) per week. Error bars (±95% CI) for individual time points may be obscured by the point.**





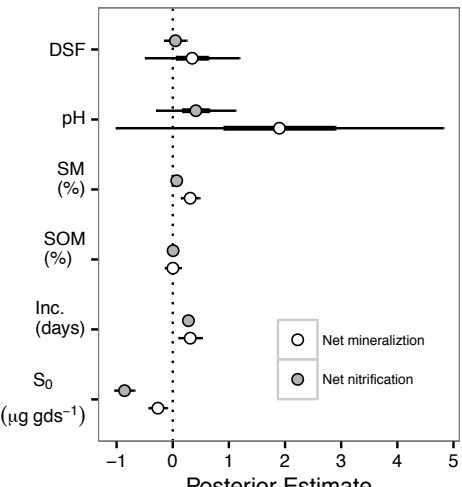

**Figure 8. Estimates of parameter posterior mean effects with bars showing 95% credible intervals (CI) for net mineralization and net nitrification rates. Parameters are days since fire (DSF), soil pH, soil moisture (SM), soil organic matter (SOM), the incubation length (Inc.), and initial substrate availability ($S_0$) per gram of dry soil (gds).**



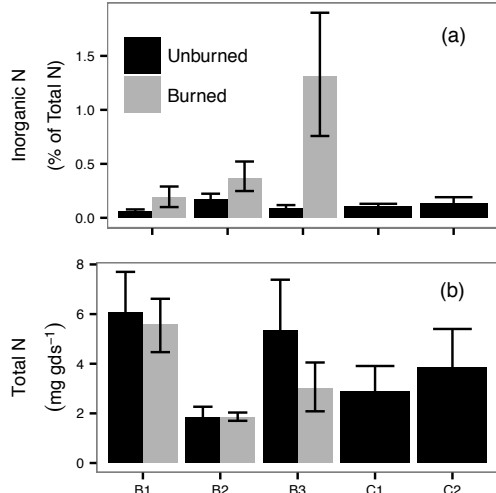

**Figure 9. Bar chart showing the pulse of inorganic N relative to the total N pool size. Panel (a) shows the mean percent of inorganic N (i.e. $NH_4^+$ and $NO_3^-$) in soil under in unburned and burned periods. Panel (b) shows the mean total N soil content in burned and unburned periods, reported per gram of dry soil (gds). Error bars are ±95% CI.**