# Peer review of "Plant controls on post-fire nitrogen availability in a pine savanna"

_Biogeosciences, 2016_

## Referee Comment (RC1) · Anonymous Referee #1 · 12 Sep 2016

General comments

The study presented in this paper investigates soil nitrogen availability in longleaf pine savanna after prescribed burning. This piece of work fits within the scope of Biogeosciences as it studies nitrogen (N) pool sizes and transformations in an open canopy savanna-like ecosystem. The paper presents a novel dataset of weekly data points spanning across a pre- and post-fire period of nine weeks. The description of the study site and scientific methods applied are clearly articulated.

The authors could improve model description for pool sizes and cycling rates by including more details. The paper is well written with a logic structure and concisely summarised in the abstract.

However, in my opinion, the results do not sufficiently support the interpretations in

the discussion, as the chosen setup of study sites does not seem adequate. Firstly, the aim of this study was to present data about the effects of prescribed fire on soil N dynamics; yet, one of the three treatment sites (B2) was affected by wildfire and had a shortened fire return interval compared to the other two sites. Secondly, the two sites affected by prescribed fire had very different responses to fire in terms of vegetation re-sprouting and different standing biomass stocks prior to fire. While the authors related the differences in magnitude of the mineral N pulse to these site differences the number of independent sites (N=2, with three replicate soil cores per site and week) seems too small to support the overall conclusion proposed in the paper - that plant uptake regulates post-fire N availability; especially given the high variance within site pre- and post-burn data and between sites.

Specific comments

The authors may consider revising Figure 1 as the schematic illustration of the paired-core sampling design is not readily understood. For example, it is unclear what the single circle below week 9 represents, is it the last sample for the measurement of pool size? It might be better to depict paired-soil cores for all nine weeks or omit the figure altogether as the sampling design is sufficiently explained in section 2.2.

Authorities for plant species names should be included when species are mentioned for the first time.

In the methods section, the description of the Bayesian hierarchical models would benefit from including more details, specifically: - Site effects (intercepts for B1-B3) should be reported - Did the authors standardise the coefficients? - What is the underlying distribution for $\beta 0i,j$? - Should the formula in 3b have a minus before $\beta 6iy0i,j$ as the initial concentration is subtracted from the incubated concentration? - Using the rjags package, how many chains and iterations were run? How was convergence tested? - Does $\sigma \sim$ unif(0,100) relate to both models or just the cycling rates model?

On page 15 (line 19), please state how soon following fire vegetation re-sprouted in

sites B1 and B2.

On page 18 (line 4), should it read "...and sharp increases in soil temperature with depth..." instead of decline?

In the discussion on page 18 (line 27) authors refer to the preference of plants in pine savanna for uptake of ammonium. It would be good to include a reference confirming this statement about uptake preference in this ecosystem as the authors argue that plant preference for ammonium uptake could explain the relatively large nitrate pool sizes relative to ammonium.

Temperature is an important influential factor on N transformation processes (MIN, NIT) and soil temperatures might change after fire due to the blackened surface promoting increased heat absorption. The authors could discuss whether they consider soil temperature to have an effect on N transformation processes in the context of their study.

Technical corrections

Page 5, line 17: delete 'in'. Page 9, line 1: correct the word 'through'. Page 10, line 23: correct reference to figures to '3-4' instead of '2-3'. Page 11, line 7: correct figure number in brackets to Figure 5. Page 17, line 9: delete first 'ash' in sentence.

---

## Referee Comment (RC2) · Anonymous Referee #2 · 9 Oct 2016

General Comments: The main objective of this research was to assess potential mechanisms responsible for post-fire variations in nitrogen (N) availability and pools in a longleaf pine savanna. The manuscript provides weekly measurements of N over a 9-week period (several weeks pre- and post-fire) during the growing season at five sites at Ft Bragg, NC. This allowed detection of rapid but ephemeral pulses in N availability post-fire. The subject matter appears appropriate for Biogeosciences. The paper is relatively well-written and thought out. However, background information provided in Introduction needs additional citation (e.g., Pg 3 lines 17-18, 21-22; Pg 4 lines 1-2, 6, 7-9, 9-11), and there is a lack of detail in the Methodology (see specific comments below). As written, the sampling design appears a bit weak, and the choice of sites and inclusion of the wildfire site and exclusion an outlier control site need more justification. The use of the Bayesian model to account for pre-fire heterogeneity of N

values appear to be a useful technique. The authors claim that decreased vegetation cover post-fire could be a major factor driving increases in NH4+ availability, but no vegetation data are provided, although these data apparently exist (cited in Discussion as 'unpublished data'). Inclusion of vegetation data (at least in some form) would substantially strengthen the paper.

Specific comments. Pg 2, Line 24: This phrase is confusing and needs clarification: "localized studies with ecosystem-specific temporal data resolution" Pg 5, line 6: Please provide examples of the Ericaceous species Pg 5, lines 10-15: It would be useful to know more about the burn regime of the study sites, especially since one of the fires was a wildfire. When are prescribed fires normally set? At the same time as the wildfire? Was the wildfire similar in severity and timing as the prescribed fires? Pg 5, lines 15-19: It's unclear what is meant by "historical burn characteristics." The information in parentheses makes it seem like this means that all the sites burned under similar conditions, but the last part of the sentence makes it seem like only the return interval was the same. Pg 5, line 20: The grouping of the wildfire with other fire sites needs more justification. The authors go to great length to explain in previous sentences why the three 'burned' sites were chosen based on their similarities and then seem to gloss over the grouping of this wildfire with other fire sites despite the fact that it burned at a totally different burn interval and likely under very different conditions. Pg 5, line 22-24: The removal of the third control site from further analyses is questionable. Were these differences among control sites unknown prior to sampling, or were they only discovered after sampling? Is there nothing that can be gleaned from information on this site? How representative of the area is this site? Pg 5, line 23: Five sites is pretty small sample size, especially since the wildfire site might not represent the prescribed fire sites. If the sites are not considered replicates, and instead the cores within a site are the replicates, this should be clarified. Pg 5, line 26: How big was the sampling area? Pg 5, line 27-28: The description of sampling above the 'ecotone' needs clarification. Was this just to avoid being in the 'extemes' of either upland or lowland? Pg 6, lines 1-2: The vegetation sampling needs to be described in more detail. Exactly how was

this done? Given that the vegetation link to N availability is a big part of this study's conclusion, why arent' these data included or at least described in more detail? Pg 6, line 3: This sentence should be moved to where the fire regimes are being described. Pg 6, lines 4-7: What temperatures were recorded? Where in the burns were these pyrometers installed? How far apart from each other? How high off the ground? Pg 6, lines 6-7: Are there other surrogates of fire intensity that could be used to assess this wildfire site? Canopy mortality? Depth of residual organic layer? Char cover? Pg 6, line 13: How far apart were cores? "Adjacent to each other" is vague. Pg 6, line 19: "throughout the growing season" makes it seem like samples were collected over a much longer period. Longleaf probably grow for many weeks (much more than 9 weeks) in North Carolina. Pg 6, line 21: "until they were analyzed" is vague. How long were soils typically stored frozen before analysis? Pg 7, line 21: What is the date referring to? Pg 8, line 24: What is SOM? Soil organic matter? From the description in previous sections, no organic layer develops in this system. Please clarify. Pg 10, line 11-12: The vegetation description should be expanded to show how the communities varied across sites. What functional types are the three species listed? Grasses, forbs, shrubs? Pg 10, line 17-18: This sentence seems out of place since no description of fuel load or moisture across the sites is given in this manuscript. Pg 14, line 16-17: The link to plant communities would be strengthened if there were more detailed plant data included in the study. As written, there's no way to assess whether N availability co-varied with plant abundance. Apparently these data exist (Pg 15, lines 18 and 21), so why aren't they included? Pg 17, this paragraph on 15N is way too long and hard to follow. Please simplify and condense, or break into a couple of paragraphs. Pg 18, lines 25-28: Inclusion of the vegetation data would substantially strengthen this statement. Figure 1 is a bit confusing. The text description could use more detail for clarification.

Technical Corrections: Throughout: "Southeastern" is sometimes one word and sometimes two words.

---

## Author Comment (AC2) · 15 Oct 2016

General Comments: The main objective of this research was to assess potential mechanisms responsible for post-fire variations in nitrogen (N) availability and pools in a longleaf pine savanna. The manuscript provides weekly measurements of N over a 9- week period (several weeks pre- and post-fire) during the growing season at five sites at Ft Bragg, NC. This allowed detection of rapid but ephemeral pulses in N availability post-fire. The subject matter appears appropriate for Biogeosciences. The paper is relatively well-written and thought out. However, background information provided in Introduction needs additional citation (e.g., Pg 3 lines 17-18, 21-22; Pg 4 lines 1-2, 6, 7-9, 9-11), and there is a lack of detail in the Methodology (see specific comments below). As written, the sampling design appears a bit weak, and the choice of sites and inclusion of the wildfire site and exclusion an outlier control site need more justification. The use of the Bayesian model to account for pre-fire heterogeneity of N values appear to be a useful technique. The authors claim that decreased vegetation cover post-fire could be a major factor driving increases in NH4+ availability, but no vegetation data are provided, although these data apparently exist (cited in Discussion as 'unpublished data'). Inclusion of vegetation data (at least in some form) would substantially strengthen the paper.

> **Thank you for taking the time to review our manuscript. We address your individual comments below. Our responses to your suggestions are bolded and indented for visual clarity.**

Specific comments. Pg 2, Line 24: This phrase is confusing and needs clarification: "localized studies with ecosystem-specific temporal data resolution"

> **This has been changed to "This suggests a need for localized studies with temporal data resolution appropriate for that ecosystem to evaluate the mechanism behind changes in soil N availability following fire."**

Pg 5, line 6: Please provide examples of the Ericaceous species

> **This has been amended to read "…dominated by Ericaceous species, including *Vaccinium formosum* Andrews, *V. fuscatum* Aiton, *V. tenellum* Aiton, *Lyonia lucida* (Lam.) K. Koch, *L. mariana* (L.) D. Don, and *Gaylussacia frondosa* (L.) Torr. and Gray ex. Torr. …"**
>
> Pg 5, lines 10-15: It would be useful to know more about the burn regime of the study sites, especially since one of the fires was a wildfire. When are prescribed fires normally

set? At the same time as the wild-fire? Was the wildfire similar in severity and timing as the prescribed fires?

**We did not measure burn temperature in the wildfire site (B2, see p10 lines 8-11), but observationally, the burn in B2 was comparable to prescribed burns in B1 and B3. See p10 lines 13-16.**

**On P5 lines 11 we added "Prescribed burns have occurred primarily during the growing seasons, when wildfires also occur. To keep the prescribed burns under control, they are performed as low-intensity backing fires (ignited along a road or other fire break and allowed to burn into the wind)."**

Pg 5, lines 15-19: It's unclear what is meant by "historical burn characteristics." The information in parentheses makes it seem like this means that all the sites burned under similar conditions, but the last part of the sentence makes it seem like only the return interval was the same.

**Prescribed fires of Sites on a 3-year burn rotation are staggered such that not *every* site burns on the same year (for example, some sites burn on years 1,4,7,10; other sites burn on years 2,5,8,11). We wanted to target sites that were on the same burn rotation and that burned in the same years. To clarify this, we added "the same" on page 5 lines 17-18 as follow: "…we limited the number of burned sites to those with similar burn histories (i.e. all on the same 3-year burn rotation). …"**

Pg 5, line 20: The grouping of the wildfire with other fire sites needs more justification. The authors go to great length to explain in previous sentences why the three 'burned' sites were chosen based on their similarities and then seem to gloss over the grouping of this wildfire with other fire sites despite the fact that it burned at a totally different burn interval and likely under very different conditions.

**Indeed the wildfire burned out of rotation, however the climatic and weather conditions at the time of all burns was similar (ie all fires occurred in a 5 day period in July). In this system, prescribed fires are employed to mimic historically naturally occurring wildfires, which occurred on a 1-3 fire return interval (Frost 1998; mean fire return interval = 2.2 years; Stambaugh et al 2011). As such, the wildfire that unexpectedly occurred, did so within the range of both wild- and prescribed-fire intervals observed in this system. Additionally, after 40 years of either 2- or 3-year fire intervals, there was no statistically significant difference reported in the % cover of any understory plant group (tree seedlings, shrubs, vines, graminoids, forbs, or ferns and mosses; Brockway & Lewis 1997). Consequently, we do not anticipate substantial differences in biomass accumulation after one year of a shortened fire return interval, and biomass is a strong determinant of fire intensity. We will amend the next draft of the manuscript to reflect this information (page 5 lines 21-27).**

Pg 5, line 22-24: The removal of the third control site from further analyses is questionable. Were these differences among control sites unknown prior to sampling, or were they only

discovered after sampling? Is there nothing that can be gleaned from information on this site? How representative of the area is this site?

> **There was some indication prior to sampling that the third control site differed from the other sites. For example, it had cinnamon fern (*Osmunda cinnamomea* L.), a species that generally grows in wetter areas, growing near our sampling area, while neither the other control nor treatment sites had this species. However, it was not until we examined data on soil nitrogen content that we realized how different this site was from the others. For example, its nitrogen content was at least one order of magnitude greater than all other sites, and it had substantially greater soil moisture than the other sites. This site is not necessarily anomalous of longleaf pine forests- the area is underlain by an undulating water table which gives rise to drastic micro-topographical gradients and high floristic diversity. However, because it had such substantially different initial soil conditions than the remainder of our sites, we excluded it from analyses. We suspect differences in the depth of the water table (specifically, it may be closer to the surface) between the site we removed from analysis and our remaining sites may be a cause of such differences in vegetation cover.**

Pg 5, line 23: Five sites is pretty small sample size, especially since the wildfire site might not represent the prescribed fire sites. If the sites are not considered replicates, and instead the cores within a site are the replicates, this should be clarified.

> **Sites are considered replicates.**

Pg 5, line 26: How big was the sampling area?

> **The sampling area was 1 m$^2$. This information will be added to the next draft of the manuscript.**

Pg 5, line 27-28: The description of sampling above the 'ecotone' needs clarification. Was this just to avoid being in the 'extremes' of either upland or lowland?

> **Indeed this was to avoid sampling either in the riparian wetland areas or the very dry upland areas. We state "This topographic location was chosen to minimize the effects of extremely well-drained, hydrologically disconnected (as found in the uplands) or saturated, anoxic (as found in the wetlands) soils on microbial processing. "**

Pg 6, lines 1-2: The vegetation sampling needs to be described in more detail. Exactly how was this done? Given that the vegetation link to N availability is a big part of this study's conclusion, why arent' these data included or at least described in more detail?

> **We collected vegetation cover data at the onset of the experiment, prior to any fires, to help ensure that our sites were appropriate replicates. We clarify this as follows: "At the onset of the experiment, all vegetation within sampling plots was identified**

**to species and the percent cover was estimated." However, we did not anticipate that vegetation regrowth could play a role in structuring post-fire nitrogen availability. Instead, this hypothesis arose as our data did not support other explanatory hypotheses. As a consequence, we did not track vegetation recovery following fire for this study.**

Pg 6, line 3: This sentence should be moved to where the fire regimes are being described.

**Because the complexities of the fire regime at each site necessitated substantial discussion in advance of actually listing the sites identifiers along with their burn dates, we found it difficult to relocate this sentence earlier in the manuscript. We chose not to move this information so that we could concisely provide the burn dates with the individual sites for all the burned sites we analyzed.**

Pg 6, lines 4-7: What temperatures were recorded? Where in the burns were these pyrometers installed? How far apart from each other? How high off the ground?

**Burn temperatures are reported in the results (page 11 lines 8-11). We amended the methods section to include to include information on the tag layout within the site.**

Pg 6, lines 6-7: Are there other surrogates of fire intensity that could be used to assess this wildfire site? Canopy mortality? Depth of residual organic layer? Char cover?

**These prescribed burns are low intensity and do not reach into the canopy. We described observational metrics of fire intensity in the results on page 10 lines 13-17.**

Pg 6, line 13: How far apart were cores? "Adjacent to each other" is vague.

**We collected cores within approximately 10 cm from each other, but we avoid sampling large roots. We will add this information to the manuscript.**

Pg 6, line 19: "throughout the growing season" makes it seem like samples were collected over a much longer period. Longleaf probably grow for many weeks (much more than 9 weeks) in North Carolina.

**We replaced "throughout" with "during".**

Pg 6, line 21: "until they were analyzed" is vague. How long were soils typically stored frozen before analysis?

**Soil cores were stored on ice while in the field and were moved to 4C within the same day. Cores were never frozen. We analyzed soils for nitrogen content within 48 hours of collection, as stated on page 6 line 25. Other analyses which are not time sensitive were analyzed in the fall.**

Pg 7, line 21: What is the date referring to?

**Thank you for catching this. This was an error on the part of our citation managing software. The correct citation for the monthly precipitation values has been added.**

Pg 8, line 24: What is SOM? Soil organic matter? From the description in previous sections, no organic layer develops in this system. Please clarify.

**An organic litter layer (ie an O horizon) does not develop, but the soil still has some (albeit very little) organic matter. We added a definition for SOM (soil organic matter) at its first mention (after revisions, p9 line 8).**

Pg 10, line 11-12: The vegetation description should be expanded to show how the communities varied across sites. What functional types are the three species listed? Grasses, forbs, shrubs?

**We added the functional types of the listed species.**

Pg 10, line 17-18: This sentence seems out of place since no description of fuel load or moisture across the sites is given in this manuscript.

**This information will be omitted.**

Pg 14, line 16-17: The link to plant communities would be strengthened if there were more detailed plant data included in the study. As written, there's no way to assess whether N availability co-varied with plant abundance. Apparently these data exist (Pg 15, lines 18 and 21), so why aren't they included?

**Unfortunately, we do not have data on vegetation regrowth patterns following fire that can be incorporated in this manuscript. The data that are cited as unpublished corresponded to data collected for other ongoing studies. That plant uptake might help to explain post-fire patterns of N availability arose because two other explanatory mechanisms failed to account for observed changes. As such, we present this as a hypothesis that can and should be tested. We stress that this study was not designed as a test of the hypothesis. We hope that we have been appropriately cautious about the strength of our arguments and conclusions (for example, we acknowledge limitations in our $^{15}$N analyses on page 18 lines 11-15); our aim was not to claim that our data conclude that plant uptake is solely responsible for post-fire N patterns, but, based on our results, to hypothesize that plant uptake may be another factor that should be explicitly considered in future studies.**

Pg 17, this paragraph on 15N is way too long and hard to follow. Please simplify and condense, or break into a couple of paragraphs.

**We have broken this into 3 separate paragraphs to improve the flow of this section.**

Pg 18, lines 25-28: Inclusion of the vegetation data would substantially strengthen this statement.

**We are unable to provide vegetation regrowth data because this experiment was designed to test how microbial cycling and ash deposition contributed to N availability.**

Figure 1 is a bit confusing. The text description could use more detail for clarification.

**Comments we received on earlier drafts of this manuscript have found this figure both helpful and unhelpful. Reviewer 1 here commented that the figure was redundant to the information provided in the text and may not be necessary, but also indicated some uncertainty about illustrations in the figure. As a consequence, we are hesitant to remove the figure entirely and prefer instead to clarify it. We are attaching a revised version of Figure 1 along with this reply. We have substantially increased the text in the figure, and would appreciate your feedback on the revised version.**

Technical Corrections: Throughout: "Southeastern" is sometimes one word and some- times two words.

**We have replaced "south eastern" with "southeastern" in all cases. Thank you for catching this.**

---

## Author Response (AR1)

Dear Dr. Ibrom,

Thank you for your comments and for your discussion with the reviewers. We understand that you and Referee #2 both feel the manuscript would be strengthened by including vegetation data to support the hypothesis that plant uptake might contribute to patterns of post-fire N availability in this system. We are concerned, however, that there has been an unintended miscommunication regarding the nature of these data. The data were not collected in any way in which they can be leveraged to empirically test this hypothesis. Even without vegetation data, however, we feel strongly that the manuscript remains an interesting and important contribution to the scientific literature. To date, the primary mechanisms put forward in the literature to explain patterns of post-fire nitrogen pulses are mineral inputs from ash and increases in microbial activity, although neither of these have been rigorously tested. Our study makes two clear advances – first we introduce a Bayesian analysis that better incorporates spatial nutrient heterogeneity into field measures of net mineralization rates. This allows us to demonstrate increases in N availability in our system. Secondly, we are able to refute both of the previously proposed mechanisms to which nitrogen pulses have been attributed. This implies that some unaccounted-for biogeochemical mechanism must be involved in regulating nutrient availability following fire disturbance. We view changes in plant uptake as the most parsimonious explanation for this pattern, but our primary goal was to propose this mechanism to the scientific community, encouraging studies that directly test alternative mechanisms of nutrient dynamics following disturbances and particularly those which examine nutrient sinks in vegetation.

We fear that including the vegetation data will weaken the manuscript by drawing attention to the methodological incompatibility of the vegetation and N availability data. Put simply, the unpublished data we cite in this manuscript come from a single time point collected two years prior to the data presented in this study. Because of this, we believe the vegetation data are incompatible with the data we present here; we cited them in the discussion to draw attention to the field-level implications of vegetation nutrient uptake and provide evidence that this proposed mechanism could be plausible. However, our intention was not to imply that our data provide direct support for this hypothesis. Of course it remains entirely possible that plant-uptake does *not* help to resolve patterns of post-fire nutrient availability, or that a different mechanism plays a larger role, as you suggest, but we were unable to identify other plausible mechanisms for this ecosystem.

We propose some organizational changes to that, we hope, clarify the distinction between which hypotheses we tested versus what we encourage future research to target. In particular, we would like to change the title of our manuscript to *The contributions of microbial activity and ash deposition to post-fire nitrogen availability in a pine savanna*. We re-organized the listed hypotheses in the abstract to differentiate the two we tested from the third untested hypotheses that we propose. Finally, in the Conclusions, we explicitly caution against drawing conclusions from our proposed plant uptake hypothesis because it remains untested.

Again, we very much appreciate the time you have invested in this manuscript. Below we attach a line-by-line responses to the reviewer comments as well as a marked up version of the manuscript tracking the changes we have made.

Sincerely,

Cari Ficken

Line Edits for RC1

The authors could improve model description for pool sizes and cycling rates by including more details. The paper is well written with a logic structure and concisely summarised in the abstract.

> **The model description will be improved based on your specific comments. See below for details.**

However, in my opinion, the results do not sufficiently support the interpretations in the discussion, as the chosen setup of study sites does not seem adequate. Firstly, the aim of this study was to present data about the effects of prescribed fire on soil N dynamics; yet, one of the three treatment sites (B2) was affected by wildfire and had a shortened fire return interval compared to the other two sites. Secondly, the two sites affected by prescribed fire had very different responses to fire in terms of vegetation re- sprouting and different standing biomass stocks prior to fire. While the authors related the differences in magnitude of the mineral N pulse to these site differences the number of independent sites (N=2, with three replicate soil cores per site and week) seems too small to support the overall conclusion proposed in the paper - that plant uptake regulates post-fire N availability; especially given the high variance within site pre- and post-burn data and between sites.

> **We believe there are two important points to consider here. The first relates to the sample size and the effects of prescribed vs wildfire. In this system, prescribed fires are employed to mimic historically naturally occurring wildfires, which occurred on a 1-3 fire return interval (Frost 1998; mean fire return interval = 2.2 years; Stambaugh et al 2011). As such, the wildfire that unexpectedly occurred, did so within the range of both wild- and prescribed-fire intervals observed in this system. After 40 years of either 2- or 3-year fire intervals, there was no statistically significant difference reported in the % cover of any understory plant group (tree seedlings, shrubs, vines, graminoids, forbs, or ferns and mosses; Brockway & Lewis 1997). Consequently, we do**

not anticipate substantial differences in biomass accumulation after one year of a shorted fire return interval, and biomass is a strong determinant of fire intensity, and we will amend the next draft of the manuscript to reflect this information as follows (page 5 line 28 – p6 lines 1-4):

*The site that burned prematurely due to a wildfire was grouped with other burned sites, despite its shortened fire return interval (one year) relative to the other burned sites (three years). Previous work has found no significant difference in vegetation cover after 40 years of management with either a 2- or 3-year burn interval (Brockway and Lewis, 1997); because biomass is a strong determinant of fire intensity, we did not anticipate that a site experiencing a shortened burn regime for one year would have substantial effects on fire dynamics.*

Nevertheless, N=3 sites remains a small sample size and certainly limits our ability to draw conclusions across ecosystems. We will be cautious about the strength of our claims in the next draft of this manuscript. If there are particular sentences in which you feel we have stretched the applicability of these findings, we would appreciate your direction to them.

Secondly, we agree that it is premature to draw conclusions that plant uptake is solely responsible for post-fire N availability dynamics. Rather than claiming that our data conclude this, we believe our data disprove two alternative hypotheses- an increase in microbial processing and ash deposition- that could independently account for the observed patterns. Instead, we hypothesise that the role of plant N uptake is another factor that should be explicitly considered in future studies. We believe we have been appropriately cautious as to the strength of our arguments and conclusions. For example, we acknowledge limitations in our $^{15}$N analyses on page 19 lines 3-7:

*Given the uncertainties surrounding the redistribution of surface inputs down the soil profile, we cannot conclusively rule out the potential to surface additions to contribute to the observed $NH_4^+$ pulse. Nevertheless, considering the unrealistic mass of ash-N needed to be deposited onto surface soils to account for our measured shifts in $\delta^{15}N$, we conclude that ash inputs are unlikely to fully account for the increase in measured soil inorganic N availability.*

And on page 19, lines 15-20, we acknowledge that it would be inappropriate to conclude that post-fire increases in $NH_4^+$ is solely driven by changes in plant sink strength:

*If fire damage temporarily halted or slowed the plant uptake of inorganic N, we would expect to see an accumulation of soil N if microbial immobilization did not increase*

*sufficiently to deplete the pool. However, N accumulating in excess of demand can only partly explain observed increases in inorganic N availability, since the pulse of N we detected following fire was many times greater than what was produced by net mineralization and net nitrification. Nevertheless, a change in plant sink strength may have contributed to post-fire $NH_4^+$ pulse.*

**Brockway, DG and CE Lewis. 1997.** *Long-term effects of dormant-season prescribed fire on plant community diversity, structure and productivity in a longleaf pine wiregrass ecosystem*. **Forest Ecology & Management (96): 167-183.**

**Frost, C.C. (1998). Presettlement fire frequency regimes of the United States: A first approximation. Proceedings 20th Tall Timbers Fire Ecology Conference: Fire in Ecosystem Management: Shifting the Paradigm from Suppression to Prescription, Boise, ID (eds T.L. Pruden & L.A. Brennan), pp. 70–81. Tall Timbers Research, Inc.,Tallahassee, FL.**

**Stambaugh, MC, RP Guyette, and JM Marschall. 2011.** *Longleaf pine (*Pinus palustris *Mill.) fire scars reveal new details of frequent fire regime*. **J. Veg. Science (22): 1094-1104.**

Specific comments

The authors may consider revising Figure 1 as the schematic illustration of the paired- core sampling design is not readily understood. For example, it is unclear what the single circle below week 9 represents, is it the last sample for the measurement of pool size? It might be better to depict paired-soil cores for all nine weeks or omit the figure altogether as the sampling design is sufficiently explained in section 2.2.

**The circle (core) below week 9 was meant to indicate that the incubating core from the final week's set was collected one week 10, since it was installed on week 9 and incubated for a week. This is consistent with how the other core sets were treated (the dashed circle depicts the core collection timing). We have received comments both that this figure is helpful and unhelpful and have opted to keep the figure, but expand its text to increase clarity.**

Authorities for plant species names should be included when species are mentioned for the first time.

**Authorities for plant species have been added.**

In the methods section, the description of the Bayesian hierarchical models would benefit from including more details, specifically: -

Site effects (intercepts for B1-B3) should be reported –

**We have included significant site effects in the text (see below for example). For visual clarity (because site effects were substantial greater than environmental effects), we have omitted them from figures. This omission note is also added to figure legends.**

**Page 12 lines 9-13:** *Site effects ($\beta_0$) had the strongest overall effect on $NH_4^+$ pool sizes, although this effect was not significant at C1. In burned sites B1-3, $\beta_0$ was -26.9 (95% credible interval (CI) = -46.11– -6.57), -22.03 (95% CI = -38.91– -4.78), and -24.16 (95% CI = -41.78– -6.12), respectively. At C2, $\beta_0$ was -23.23 (95% CI = -40.38–-5.69).*

**Page 12 lines 22-23:** *Site effects on $NO_3^-$ pool sizes much weaker than for $NH_4^+$ and were only significant at B1 ($\beta_0$ = -4.81, 95% CI = -9.46 – -0.08).*

Did the authors standardise the coefficients? –

**Coefficients were not standardised prior to analysis.**

What is the underlying distribution for $\beta 0i,j$? –

**$\beta 0i,j$ has a normal underlying distribution. This information has been added to the text on line 22-23 (page 10):** *All predictors, including random site site effects, were modelled with normally distributed, uninformative priors.*

Should the formula in 3b have a minus before $\beta 6iy0i,j$ as the initial concentration is subtracted from the incubated concentration? –

**The effects of initial concentrations are reflected in the negative posterior estimates of $S_0$ (Fig 8).**

Using the rjags package, how many chains and iterations were run?

**Three chains were run with 200,000 iterations after a 100,000-iteration burn-in period. This information will be included in the Methods section (page 10, lines 23-24).**

How was convergence tested? –

**Convergence was tested by examining chain density and trace plots to ensure proper chain mixing, and by calculating the Gelman-Rubin diagnostics using the gelman.diag() function in the coda package to ensure the scale reduction factors for each predictor was <1.05. This information has been added to the methods section (p 10 line 25 – p11**

**line 1.**

Does σ ~ unif(0,100) relate to both models or just the cycling rates model?

**This relates to both models and has been included in the set of equations describing the pool size model as well (p9 line 23).**

On page 15 (line 19), please state how soon following fire vegetation re-sprouted in sites B1 and B2.

**The information was amended as follow: Page 16 (line 20)** *"We noticed that B1 and B2 exhibited rapid vegetation resprouting following fire, while regrowth in B3 was patchy. Vegetation began resprouting in B1 and B2 six days after fire, but not until 18 days after fire in B3 (C. Ficken, unpublished data)."*

On page 18 (line 4), should it read "...and sharp increases in soil temperature with depth. . ." instead of decline?

**Yes, it should read "increases", and we changed the word. Thank you for catching this.**

In the discussion on page 18 (line 27) authors refer to the preference of plants in pine savanna for uptake of ammonium. It would be good to include a reference confirming this statement about uptake preference in this ecosystem as the authors argue that plant preference for ammonium uptake could explain the relatively large nitrate pool sizes relative to ammonium.

**As far as we know, there are no studies explicitly documenting the nitrate vs ammonium preference for species inhabiting pine savannas, but preference for one N form is likely the result of multiple drivers, including enhanced uptake of the dominant N source (e.g. Kronzucker et al 1997; Houlton et al 2007; Wang and Mack 2011). Because nitrification rates are low at low pH, acidic soils often have greater ammonium availability than nitrate. If the availability of each nitrogen form is one component of preference, we expected that plants inhabiting the acidic soils of our study system to take up relatively more ammonium than nitrate. Here, we draw an analogy between seasonal patterns of N availability in longleaf pine savanna (high soil ammonium and low nitrate during the winter, but low ammonium and high nitrate during the growing season; Christensen 1977), and seasonal patterns of N concentrations in northeastern US streams (winter maxima when terrestrial plant N uptake is low; Vitousek 1977). To clarify this, we added references explaining the drivers of plant N uptake patterns and resulting environmental availability, and amended the manuscript as follows (Page 20 lines 5:9):**

*Preference for $NH_4^+$ by plants inhabiting acidic soils, where nitrification is limited by low pH and $NO_3^-$ availability is consequently low (Ste-Marie and Paré, 1999;Houlton et al., 2007;Wang and Macko, 2011;Kronzucker et al., 1997), could help to explain the relatively large pool sizes of $NO_3^-$ relative to $NH_4^+$ during the growing season (Vitousek, 1977), and this pattern is consistent with previous seasonal trends in a longleaf pine savannas (Christensen, 1977).*

Temperature is an important influential factor on N transformation processes (MIN, NIT) and soil temperatures might change after fire due to the blackened surface promoting increased heat absorption. The authors could discuss whether they consider soil temperature to have an effect on N transformation processes in the context of their study.

**Indeed, temperature is an influential factor for N transformations. To reflect this, we added the following sentence on page 17 (line 10), and included two citations for readers interested in learning more. We limited the discussion of this driver, however, since we did not observe increases in cycling rates following fire.**

*Soil surface blackening after fire may increase soil temperature and stimulate immediate and prolonged N transformations after fire (Booth et al., 2005;Ojima et al., 1994).*

Technical corrections Page 5, line 17: delete 'in'. Page 9, line 1: correct the word 'through'. Page 10, line 23: correct reference to figures to '3-4' instead of '2-3'. Page 11, line 7: correct figure number in brackets to Figure 5. Page 17, line 9: delete first 'ash' in sentence.

**These technical corrections have been made, and we appreciate your careful review of our manuscript.**

Line Edits for RC2

Specific comments. Pg 2, Line 24: This phrase is confusing and needs clarification: "localized studies with ecosystem-specific temporal data resolution"

**This has been changed to "This suggests a need for localized studies with temporal data resolution appropriate for that ecosystem to evaluate the mechanism behind changes in soil N availability following fire."**

Pg 5, line 6: Please provide examples of the Ericaceous species

**This has been amended to read "…dominated by Ericaceous species, including *Vaccinium formosum* Andrews*, V. fuscatum* Aiton, *V. tenellum* Aiton, *Lyonia lucida* (Lam.) K. Koch, *L. mariana* (L.) D. Don, and *Gaylussacia frondosa* (L.) Torr. and Gray ex. Torr. …"**

Pg 5, lines 10-15: It would be useful to know more about the burn regime of the study sites, especially since one of the fires was a wildfire. When are prescribed fires normally set? At the same time as the wild-fire? Was the wildfire similar in severity and timing as the prescribed fires?

**We did not measure burn temperature in the wildfire site (B2, see p6 lines 14-19), but observationally, the burn in B2 was comparable to prescribed burns in B1 and B3. See p11 lines 8-11.**

**On P5 lines 12-16 we added "Prescribed burns are set primarily during the growing seasons, when wildfires also occur. To maintain control of the prescribed burns, they are performed as low-intensity backing fires (ignited along a road or other fire break and allowed to burn into the wind). This fire return interval mimics the historical fire return interval of 1 to 3 years (Frost, 1998), with burns occurring on average every 2.2 years (Stambaugh et al., 2011)."**

Pg 5, lines 15-19: It's unclear what is meant by "historical burn characteristics." The information in parentheses makes it seem like this means that all the sites burned under similar conditions, but the last part of the sentence makes it seem like only the return interval was the same.

**Prescribed fires of Sites on a 3-year burn rotation are staggered such that not *every* site burns on the same year (for example, some sites burn on years 1,4,7,10; other sites burn on years 2,5,8,11). We wanted to target sites that were on the same burn rotation and that burned in the same years. To clarify this, we added "the same" on page 5 line 25 as follow: "…we limited the number of burned sites to those with similar burn histories (i.e. all on the same 3-year burn rotation). …"**

Pg 5, line 20: The grouping of the wildfire with other fire sites needs more justification. The authors go to great length to explain in previous sentences why the three 'burned' sites were chosen based on their similarities and then seem to gloss over the grouping of this wildfire with other fire sites despite the fact that it burned at a totally different burn interval and likely under very different conditions.

**Indeed the wildfire burned out of rotation, however the climatic and weather conditions at the time of all burns was similar (ie all fires occurred in a 5 day period in July). In this system, prescribed fires are employed to mimic historically naturally occurring wildfires, which occurred on a 1-3 fire return interval (Frost 1998; mean fire return interval = 2.2 years; Stambaugh et al 2011). As such, the wildfire that unexpectedly occurred, did so within the range of both wild- and prescribed-fire intervals observed in this system. Additionally, after 40 years of either 2- or 3-year fire intervals, there was no statistically significant difference reported in the % cover of any understory plant group (tree seedlings, shrubs, vines, graminoids, forbs, or ferns and mosses; Brockway & Lewis 1997). Consequently, we do not anticipate substantial differences in biomass accumulation after one year of a shortened fire return interval, and biomass is a strong determinant of fire intensity. We will amend the next draft of the manuscript to reflect this information (page 5 line 28 to page 6 lines 1-4).**

Pg 5, line 22-24: The removal of the third control site from further analyses is questionable. Were these differences among control sites unknown prior to sampling, or were they only discovered after sampling? Is there nothing that can be gleaned from information on this site? How representative of the area is this site?

**There was some indication prior to sampling that the third control site differed from the other sites. For example, it had cinnamon fern (*Osmunda cinnamomea* L.), a species that generally grows in wetter areas, growing near our sampling area, while neither the other control nor treatment sites had this species. However, it was not until we examined data on soil nitrogen content that we realized how different this site was from the others. For example, its nitrogen content was at least one order of magnitude greater than all other sites, and it had substantially greater soil moisture than the other sites. This site is not necessarily anomalous of longleaf pine forests- the area is underlain by an undulating water table which gives rise to drastic micro-topographical gradients and high floristic diversity. However, because it had such substantially different initial soil conditions than the remainder of our sites, we excluded it from analyses. We suspect differences in the depth of the water table (specifically, it may be closer to the surface) between the site we removed from analysis and our remaining sites may be a cause of such differences in vegetation cover.**

Pg 5, line 23: Five sites is pretty small sample size, especially since the wildfire site might not represent the prescribed fire sites. If the sites are not considered replicates, and instead the cores within a site are the replicates, this should be clarified.

**Sites are considered replicates.**

Pg 5, line 26: How big was the sampling area?

**The sampling area was 1 m$^2$. This information will be added to the next draft of the manuscript (p6 line 9)**

Pg 5, line 27-28: The description of sampling above the 'ecotone' needs clarification. Was this just to avoid being in the 'extremes' of either upland or lowland?

**Indeed this was to avoid sampling either in the riparian wetland areas or the very dry upland areas. We state (p6 line 10) "This topographic location was chosen to minimize the effects of extremely well-drained, hydrologically disconnected (as found in the uplands) or saturated, anoxic (as found in the wetlands) soils on microbial processing."**

Pg 6, lines 1-2: The vegetation sampling needs to be described in more detail. Exactly how was this done? Given that the vegetation link to N availability is a big part of this study's conclusion, why arent' these data included or at least described in more detail?

**We collected vegetation cover data at the onset of the experiment, prior to any fires, to help ensure that our sites were appropriate replicates. We clarify this as follows (p6 lines 12-13): "At the onset of the experiment, all vegetation within sampling plots was identified to species and the percent cover was estimated." However, we did not anticipate that vegetation regrowth could play a role in structuring post-fire nitrogen availability. Instead, this hypothesis arose as our data did not support other explanatory hypotheses. As a consequence, we did not track vegetation recovery following fire for this study.**

Pg 6, line 3: This sentence should be moved to where the fire regimes are being described.

**Because the complexities of the fire regime at each site necessitated substantial discussion in advance of actually listing the sites identifiers along with their burn dates, we found it difficult to relocate this sentence earlier in the manuscript. We chose not to move this information so that we could concisely provide the burn dates with the individual sites for all the burned sites we analyzed.**

Pg 6, lines 4-7: What temperatures were recorded? Where in the burns were these pyrometers installed? How far apart from each other? How high off the ground?

**Burn temperatures are reported in the results (page 11 lines 10-12). We amended the methods section to include to include information on the tag layout within the site.**

Pg 6, lines 6-7: Are there other surrogates of fire intensity that could be used to assess this wildfire site? Canopy mortality? Depth of residual organic layer? Char cover?

**These prescribed burns are low intensity and do not reach into the canopy. We described observational metrics of fire intensity in the results on page 11 lines 8-10.**

Pg 6, line 13: How far apart were cores? "Adjacent to each other" is vague.

**We collected cores within approximately 10 cm from each other, but we avoid sampling large roots. We will add this information to the manuscript (p 6 line 25).**

Pg 6, line 19: "throughout the growing season" makes it seem like samples were collected over a much longer period. Longleaf probably grow for many weeks (much more than 9 weeks) in North Carolina.

**We replaced "throughout" with "during".**

Pg 6, line 21: "until they were analyzed" is vague. How long were soils typically stored frozen before analysis?

**Soil cores were stored on ice while in the field and were moved to 4C within the same day. Cores were never frozen. We analyzed soils for nitrogen content within 48 hours of collection, as stated on page 7 line 10. Other analyses which are not time sensitive were analyzed in the fall.**

Pg 7, line 21: What is the date referring to?

**Thank you for catching this. This was an error on the part of our citation managing software. The correct citation for the monthly precipitation values has been added.**

Pg 8, line 24: What is SOM? Soil organic matter? From the description in previous sections, no organic layer develops in this system. Please clarify.

**An organic litter layer (ie an O horizon) does not develop, but the soil still has some (albeit very little) organic matter. We added a definition for SOM (soil organic matter) on p9 line 19).**

Pg 10, line 11-12: The vegetation description should be expanded to show how the communities varied across sites. What functional types are the three species listed? Grasses, forbs, shrubs?

**We added the functional types of the listed species.**

Pg 10, line 17-18: This sentence seems out of place since no description of fuel load or moisture across the sites is given in this manuscript.

**This information will be omitted.**

Pg 14, line 16-17: The link to plant communities would be strengthened if there were more detailed plant data included in the study. As written, there's no way to assess whether N

availability co-varied with plant abundance. Apparently these data exist (Pg 15, lines 18 and 21), so why aren't they included?

**Unfortunately, we do not have data on vegetation regrowth patterns following fire that can be incorporated in this manuscript. The data that are cited as unpublished corresponded to data collected for other studies with different temporal resolutions. Consequently, the data were not collected in any way in which they can be leveraged to empirically test this hypothesis. That plant uptake might help to explain post-fire patterns of N availability arose because our data refuted two primary mechanisms that are commonly used to explain post-fire increases in soil N availability. This implies that there is some un-accounted for biogeochemical mechanisms that influences post-fire soil N availability; we view plant uptake as a parsimonious alternative hypothesis. We stress, however, that this study was not designed as a test of the hypothesis (p22 lines 2-4). Instead, our goal was to propose it to encourage future explicit tests of this hypothesis (p1 lines 21-23 and p22 lines 4-5). We hope that we have been appropriately cautious about the strength of our arguments and conclusions (for example, we acknowledge limitations in our $^{15}$N analyses on page 19 lines 3-7); our aim was not to claim that our data conclude that plant uptake is solely responsible for post-fire N patterns, but, based on our results, to hypothesize that plant uptake may be another factor that should be explicitly considered in future studies. We have modified the title to reflect the initial goals the experiment. We have also replaced "propose" with "speculate" (e.g. p 15 line 16; p22 line 13) to emphasize that the plant uptake hypothesis has only been described, not explicitly tested, in this study.**

Pg 17, this paragraph on 15N is way too long and hard to follow. Please simplify and condense, or break into a couple of paragraphs.

**We have broken this into 3 separate paragraphs to improve the flow of this section.**

Pg 18, lines 25-28: Inclusion of the vegetation data would substantially strengthen this statement.

**We are unable to provide vegetation regrowth data because this experiment was designed to test how microbial cycling and ash deposition contributed to N availability.**

Figure 1 is a bit confusing. The text description could use more detail for clarification.

**Comments we received on earlier drafts of this manuscript have found this figure both helpful and unhelpful. Reviewer 1 here commented that the figure was redundant to**

**the information provided in the text and may not be necessary, but also indicated some uncertainty about illustrations in the figure. As a consequence, we are hesitant to remove the figure entirely and prefer instead to clarify it. We are attaching a revised version of Figure 1 along with this reply. We have substantially increased the text in the figure, and would appreciate your feedback on the revised version.**

Technical Corrections: Throughout: "Southeastern" is sometimes one word and some- times two words.

**We have replaced "south eastern" with "southeastern" in all cases. Thank you for catching this.**

[revised manuscript text omitted]

---

## Author Response (AR2)

Please describe the method used to estimate vegetation percent cover (lines 12-13, page 6).

We included more information about how we estimated percent cover. We amended this sentence to say (p6 lines 12-13):

[revised manuscript text omitted]